# MiniOpt: Reasoning to Model and Solve General Optimization Problems with Limited Resources

## Abstract

Modeling and solving optimization problems via large language models (LLMs) has attracted increasing attention recently. Although both prompt-based and learning-based methods have achieved progress, they remain limited by their reliance on large data volumes, high-quality annotations, expensive intermediate step verification, and huge computational overhead. From a data privacy perspective, a low-cost localized deployment of small-scale LLMs is of significant value. To train a small-scale LLM with excellent optimization generalization under limited resources, this paper proposes a reasoning to model and solve paradigm called MiniOpt based on reinforcement learning (RL) with verifiable reward. To reduce the demand for training data, MiniOpt adopts two-stage RL training. In the first stage the model quickly learns the model-and-solve paradigm and in the second stage it acquires strong optimization generalization ability. To reduce the cost of verifying the response of LLMs, OptReward in MiniOpt verifies the completeness of problem modeling and avoids the need for content validation. The above techniques enable the training of small-scale LLMs with strong optimization generalization ability under limited resources, thereby resulting in low inference cost for localized deployment and usage. Extensive experiments show that MiniOpt-3B exhibits strong optimization generalization across various optimization types and scenarios. For models with parameters fewer than 10B, MiniOpt-3B achieves the highest average solving accuracy (SA). For models with more than 10B parameters, MiniOpt-3B still shows competitive performance. Notably, MiniOpt-3B indicates superior SA on the hard OptMATH-Bench while only consuming 37.64% of the average output tokens required by DeepSeek-R1. The code is available at https://anonymous.4open.science/r/MiniOpt-6194.

## 1 Introduction

Optimization problems are ubiquitous in real-world scenarios, profoundly affecting diverse domains, including industrial production and transportation planning (Song et al., 2023; Li et al., 2025b). While traditional optimization solvers are efficient, their application heavily relies on expert knowledge, requiring the manual conversion of problems described in natural language into precise mathematical formulations or code, which is a process that both time-consuming and non-generalizable. The rise of LLMs has opened new pathways for the automated modeling and solving of optimization problems using natural language descriptions (Deng et al., 2024; Sun et al., 2024; Li et al., 2025a), bringing them closer to practical application scenarios. Representative work like LLMOPT (Jiang et al., 2025) or Text2Zinc (Singirikonda et al., 2025) significantly advances this field by parsing natural language descriptions into structured formulation, which are a unified general expression for optimization problems, and subsequently generating solution codes efficiently.

However, deploying such LLM-based approaches faces three critical bottlenecks. First, ensuring accurate text generations on a smaller parameter scale requires Supervised Fine-Tuning (SFT) with a large amount of high-quality training data (Wu et al., 2025; Lu et al., 2025). However, obtaining such data is very difficult, often requires a lot of time and effort, and is prone to errors (Xiao et al., 2025b). Second, it is challenging to verify whether the result generated meets the requirements (Zhai et al., 2025). The non-verifiability of this task also exposes the limitations of the learning-based

solving paradigm. It is a barrier to the accuracy of the final result. Finally, to ensure the reliability of the result, common methods include reflection or self-correction mechanisms (Jiang et al., 2025; Li et al., 2025c), or multi-agent chains (AhmadiTeshnizi et al., 2024). However, these methods significantly multiply the computational overhead. These disadvantages pose significant barriers to implementing LLMs with powerful optimization-solving capabilities in the business scenarios of small and medium-sized enterprises or even on mobile devices. At the same time, considering data privacy issues, it is significant to locally deploy small-scale LLMs with strong performance while reducing training costs, including the volume and quality of training data.

To address these problems, this paper proposes a framework called MiniOpt to train a small-scale LLM with excellent optimization generalization under limited resources. To reduce the demand for data volume, MiniOpt adopts a two-stage RL training pipeline. Before conducting RL training, an SFT warm-up is applied to enable the model to get effective rewards in the early stages of RL. Subsequently, in the first RL stage, the model rapidly acquires the model-and-solve paradigm. In the second RL stage, training is conducted on a high-quality data subset obtained through type-scenario guided data selection, yielding strong optimization generalization (Jiang et al., 2025). The RL process is training using the proposed OptGRPO algorithm, which enhances data utilization efficiency and improves the model's ability to solve complex problems. To reduce the validation cost, we introduce OptReward in MiniOpt, which will perform three tasks: validating data format, ensuring the completeness of optimization problem modeling, and verifying solution accuracy. This encourages the model to learn in accordance with the model-and-solve paradigm. The framework enables the training of small-parameter LLMs with strong optimization generalization capabilities under limited resources, making localized deployment of LLMs for optimization feasible, thereby resulting in low inference cost for localized deployment and usage. Moreover, due to the high solving efficiency of MiniOpt-3B, reducing token usage also leads to considerable economic benefits.

Building upon the aforementioned methodology, this paper conducts extensive experiments with MiniOpt-3B on 9 benchmarks across different optimization types and problem scenarios. The results demonstrate its high optimization generalization capability. Compared to baselines below 10B parameters, MiniOpt-3B achieves the best performance. When evaluated against baselines exceeding 10B parameters, MiniOpt-3B exhibits only a 1.57% lower average solving accuracy (SA) than DeepSeek-R1, 0.37% higher than GPT-5 and outperforms LLMOPT-14B by 2.13% in average SA. Notably, MiniOpt Pareto dominates other baselines in terms of both parameter scale and the SA metric. On the hard OPTMATH-Bench, MiniOpt-3B has a higher SA than DeepSeek-R1 and only consumes about 37.64% of DeepSeek-R1's token count. Furthermore, results from ablation studies indicate that RL training contributes most significantly to MiniOpt-3B.

The subsequent sections review the related work, introduce the MiniOpt framework, present experimental results and analysis, provide an in-depth discussion, and finally conclude the paper.

## 2 RELATED WORK

**LLMs for Modeling and Solving Optimization Problems.** For modeling and solving optimization problems with LLMs, there are already a variety of benchmarks (Huang et al., 2025b; AhmadiTeshnizi et al., 2024; Yang et al., 2024b). Challenging benchmarks like OptMATH (Lu et al., 2025) and NL4Opt (Huang et al., 2025a) have led to numerous studies utilizing LLMs to solve optimization problems. Prompt-based approaches such as OptiMUS (AhmadiTeshnizi et al., 2024), CoE (Xiao et al., 2024) and LEAN-LLM-OPT (Liang et al., 2025) utilizes the powerful generation capability of LLMs to generate the solver code of the optimization problem through multi-stage pipeline, without performing any post-training. Learning-based methods enhance LLMs' capabilities in modeling and solving mathematical problems. For example, LLaMoCo (Ma et al., 2024) proposes an SFT-based framework comprising a meticulously designed instruction set and a two-stage training methodology that incorporates contrastive learning warm-up followed by SFT. LLMOPT (Jiang et al., 2025) and NER4OPT (Dakle et al., 2023; Singirikonda et al., 2025) adopts a two-stage training process of modeling the optimization problems first and then solving them by generating solution code.

**Reinforcement Learning with Verifiable Reward.** While Reinforcement Learning from Human Feedback (RLHF) (Ouyang et al., 2022) plays a crucial role in post-training alignment, it suffers from high annotation costs and inherent human bias (Xiao et al., 2025b). Reinforcement Learning with Verifiable Reward (RLVR) (Lambert et al., 2024) leverages externally grounded, easily verifi-

able rewards (e.g., rule-based reward) to provide dense and structurally simple supervision (Wang et al., 2024; Xie et al., 2025; Gao et al., 2024). Its practicality is especially valuable in real-world black-box systems (Zhang et al., 2025; Xin et al., 2025; Pan et al., 2025), where verification is typically feasible only at the output stage, making RLVR a broadly applicable framework for aligning LLMs. In the field LLMs for solving optimization problems, SIRL (Chen et al., 2025) and OR-R1 (Ding et al., 2025) significantly improves the model's performance through the RLVR training paradigm.

## 3 METHODOLOGY: THE PROPOSED MINIOPT

### 3.1 OVERVIEW

This paper studies how to endow small-scale LLMs with strong optimization generalization under tight data and compute budgets. We introduce MiniOpt, whose framework is shown in Figure 1. MiniOpt introduces the reasoning process within the model-and-solve paradigm for optimization problems, the reward function and algorithm used for RL training, and the training pipeline for small-scale models. Through this framework, MiniOpt formulates the path from a natural language problem to an executable solver code as a single verifiable end-to-end task.

### 3.2 REASONING TO MODEL AND SOLVE PARADIGM

As shown in subfigure (a) in Figure 1, we introduce a reasoning to model and solve paradigm that turns a natural language optimization problem into a single verifiable objective. The paradigm is enforced by two compulsory and machine-parsable segments `<think>...</think>` and `<answer>...</answer>`.

The first segment, enclosed by `<think>...</think>`, carries all modeling contents. It performs an analysis of the problem statement, specifies a complete five-element formulation inspired by LLMOPT (Jiang et al., 2025) and determines the appropriate open-source solver. Specifically, the optimization problem can be described as minimizing the **objective** function $f(x)$ subject to the **constraints** $G(x) \leq c$, where the $x \in \mathcal{X} \subseteq \mathbb{R}^D$ is the $D$-dimensional decision **variables**, and $I = \{1, 2, \ldots, D\}$ is the index **sets**. $\mathcal{X}$ is the feasible region, and $c \in \mathbb{R}^m$ provides the upper bounds. The constants in $f(x)$ and $G(x)$ form the **parameters** set, which also includes $c$. The five-element formulation, $\mathcal{M} = ($**Variables**, **Objective**, **Constraints**, **Sets**, **Parameters**$)$ map one-to-one to the components of an optimization problem. More details about the five-element formulation are provided in Appendix C.

During the reasoning process, to enhance the capability of LLM for solving optimization problems across diverse optimization types, it will analyze the problem during the thinking (i.e., reasoning) phase and select different open-source solvers for the given problem depending on the optimization types and the characteristics of the solvers, thereby ensuring an optimized match between the problem and the solver. Solver selection is guided by a prompt, which is introduced in Appendix M.

The second segment, enclosed by `<answer>...</answer>`, converts the five-element formulation into an executable Pyomo program that models the problem, invokes the solver, solves the instance, and prints the numerical answer. By constraining responses in this manner, we collapse the action space from free-form language to a programmable artifact whose intermediate structure and final result can be deterministically parsed and verified. The `<answer>` segment must implement the blueprint as a single python code fence containing a complete Pyomo script. Because these outputs are easy to verify, the OptReward in Section 3.4.1 can score format, five-element formulation, and accuracy based on rules via an automated procedure, providing low-cost supervision. During the training process, the prompt corresponding to this paradigm is shown in Appendix N.

### 3.3 TRAINING PIPELINE OF MINIOPT

Based on the paradigm described in Section 3.2 and the key techniques introduced in Section 3.4, we propose a training pipeline that enables MiniOpt to learn from limited resource constraints while achieving powerful solving performance and strong optimization generalization. This pipeline be-

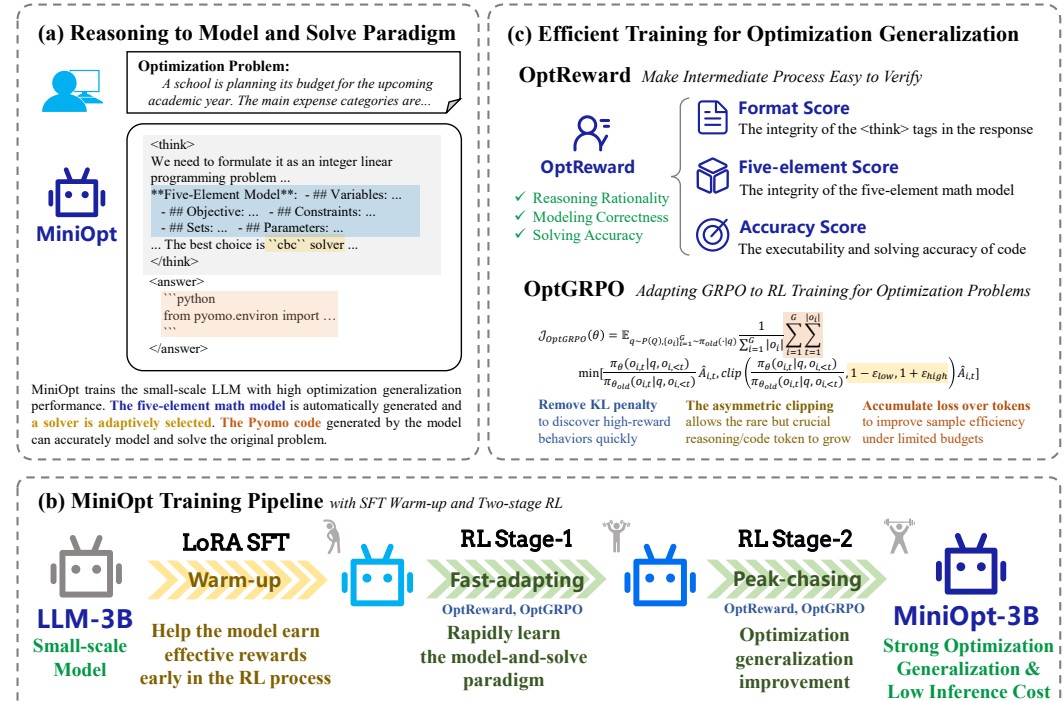

Figure 1: The framework of MiniOpt. Sub-figure (a) demonstrates the reasoning to model and solve paradigm of MiniOpt, encompassing problem modeling and solver adaptation during the thinking (i.e., reasoning) process of RL, and solution code generation in the response. Sub-figure (b) illustrates the training pipeline of MiniOpt, which involves sequential execution of SFT warm-up followed by two-stage RL. Sub-figure (c) presents the reward function OptReward and training algorithm OptGRPO used in MiniOpt's RL training.

gins with a lightweight SFT warm-up followed by two-stage RL under OptReward. The illustration of the training pipeline is shown in subfigure (b) in Figure 1.

### 3.3.1 WARM-UP BASED ON LIGHTWEIGHT SFT

Small scale LLMs struggle to produce an end-to-end, executable trajectory for complex optimization tasks, thus if a model lacks fundamental problem-solving capabilities for operations optimization, directly applying reinforcement learning may lead to sparse rewards and result in unstable training.

We therefore apply an SFT warm-up whose purpose is to provide a run-through capability, thereby allowing the model to obtain effective rewards in the early stages of RL training. This SFT warm-up does not adopt the reasoning to model and solve paradigm introduced in Section 3.2. Instead, it establishes a starting point upon which our two-stage RL can subsequently focus on paradigm acquisition and optimization generalization. The training data for warm-up are constructed from OptMATH-Train. For each instance, we prompt Qwen2.5-Coder-32B-Instruct (Hui et al., 2024) using the prompt in Appendix L to rewrite the original GurobiPy program into an equivalent Pyomo implementation and to select an appropriate open-source solver according to the detected structure. Every rewritten instance is executed and only those that compile, solve, and print the correct structure are retained. The remaining data serves as a candidate pool for the next stage of the two-stage RL training.

### 3.3.2 THE TWO-STAGE RL

After undergoing SFT warm-up, the models can conduct RL training more effectively. We employ a two-stage RL training under the same reward function (OptReward) for paradigm acquisition and optimization generalization, respectively. Both stages share the pipeline of parsing, executing,

and scoring in Section 3.2 and the OptReward of Sections 3.4.1, they differ in training data and hyperparameter setting. Algorithmically, both stages use the same OptGRPO (cf. paragraphs in Section 3.4.2).

Stage-1 aims to enable the model to acquire the reasoning to model and solve Paradigm: the model must generate a valid `<think>`/`<answer>` pair, produce the executable Pyomo code, and make a coherent solver choice. To this end we train on 1,585 relatively easy problems, so that most signal arises from the formatting and structural components of OptReward, rapidly improving executability and the ability to solve the optimization problem of natural language description.

Stage-2 focuses on optimization generalization once the paradigm is established. The training distribution shifts to the problems with diverse optimization types and problem scenario, and the optimization emphasis moves to the accuracy score, encouraging refined modeling and solving behaviors (e.g., variable/constraint formulation and solver selection).

Based on the annotations of optimization types and problem scenario tags this paper assigned to each instance in OptMATH-Train, we sample a data subset from the candidate pool mentioned in Section 3.3.1 subject to two constraints: (i) type-uniform coverage, with exactly 600 instances per type, and (ii) within each type, the scenario frequencies match the distribution in the full dataset. The resulting data serves as the training data for the second stage of RL. While the dataset of the first stage is the union of the NL4Opt (AhmadiTeshnizi et al., 2024) and ICML Competition (Yang et al., 2024b) training splits. Detailed information on the construction of the training set is provided in the Appendix E.

Such a training strategy makes efficient use of limited data and reduces training costs, and ultimately allows for strong optimization generalization of LLMs with even small parameters and limited computational resources.

## 3.4 Efficient Training for Optimization Generalization

Building upon the reasoning to model and solve paradigm mentioned in Section 3.2, we propose two key components for the RL training of MiniOpt as shown in subfigure(c) in Figure 1. An informative and easily verifiable reward function OptReward, and an improved algorithm OptGRPO builds upon GRPO (Shao et al., 2024).

### 3.4.1 OptReward: Verifiable Rewards Designed for MiniOpt

Based on the reasoning to model and solve paradigm described in section 3.2, we propose OptReward, including three automatically computed components: formatting correctness, structural sufficiency, and numerical accuracy. Each component is derived from deterministic parsing or execution of the output, It ensures the completeness of the modeling of problems but also enables verification to scale with machine time, yielding substantially lower verification costs.

**Format Score:** The format score $S_{\text{fmt}}$ validates the response format. A response must contain exactly one `<think>`...`</think>` and one `<answer>`...`</answer>` in the correct order. If all conditions hold, we assign $S_{\text{fmt}} = +1$, otherwise $S_{\text{fmt}} = -1$. If the specified format is not present in the response, we deterministically set the remaining components to their error defaults, $S_{\text{five}} = -1$ and $S_{\text{acc}} = -2$, so that the total reward immediately reaches the global minimum. This forces the model to adopt the correct response format early in training and prevents expensive evaluation of malformed samples.

**Five-element Score:** To avoid ground-truth labeling of five-element content and the bias it may introduce, we use a presence-based rule aligned with the paradigm in Section 3.2. Conditional on valid formatting, we compute the five-element score $S_{\text{five}}$. In the `<think>` segment, the model response is expected to include five labeled summaries starting with "## Sets:", "## Parameters:", "## Variables:", "## Objective:", and "## Constraints:". Each present summary adds 0.2 scores; if none is present we assign $S_{\text{five}} = -1$. This structure shaping keeps the modeling blueprint parsable.

$$S_{\text{five}} = \begin{cases} 0.2 \sum_{k=1}^{5} I_k, & \text{if } \sum_{k=1}^{5} I_k \geq 1, \\ -1, & \text{if } \sum_{k=1}^{5} I_k = 0, \end{cases} \tag{1}$$

where $I_k = 1$ if the $k$-th required element $e_k$ is present in the `<think>` segment, and $I_k = 0$ otherwise. $(e_1, \ldots, e_5) = (Sets, Parameters, Variables, Objective, Constraints)$.

**Accuracy Score:** The accuracy score is obtained by executing the Pyomo program contained in `<answer>`. Failure to extract a program or to complete execution yields $S_{\text{acc}} = -2$. When execution succeeds, we retrieve the optimal objective value $\hat{f}$ from the model output and compare it with the ground-truth value $f^\star$; if they are equal, we assign $S_{\text{acc}} = 2$, otherwise $S_{\text{acc}} = -1.5$.

$$S_{\text{acc}} = \begin{cases} +2, & \text{if execution succeeds and } \hat{f} = f^\star, \\ -1.5, & \text{if execution succeeds but } \hat{f} \neq f^\star, \\ -2, & \text{if no executable code or execution fails}. \end{cases} \tag{2}$$

Combining the components with the formatting gate yields the total OptReward as follow:

$$R = \begin{cases} -4, & \text{if } S_{\text{fmt}} = -1, \\ S_{\text{fmt}} + S_{\text{five}} + S_{\text{acc}}, & \text{if } S_{\text{fmt}} = 1. \end{cases} \tag{3}$$

By OptReward, the format score enforces the strict `<think>`/`<answer>` format, the five-element score shapes a complete modeling blueprint in the think phase, and the accuracy score certifies correctness through equality of optimal objective values, enabling low-cost verifiable RL for problems in the field of optimization.

### 3.4.2 OptGRPO: Training Small-Scale LLMs with Limited Resources

GRPO (Shao et al., 2024) is an efficient RL algorithm, which replaces the critic model with a group baseline and updates the policy at the group level so that improves training stability and efficiency. The details of the algorithm are introduced in Appendix D. In order to adapt to the designed OptReward and Two-Stage RL in this paper, taking inspiration from the previous work on the improvement of GRPO like DAPO (Yu et al., 2025), we introduces three modifications, with the goal of achieving stronger optimization generalization within a limited budget.

First, building upon the strict gating already provided by OptReward through formatting score and five-element score, we set the coefficient on the KL penalty $\beta = 0$ to remove the KL penalty to encourage greater exploration by the model. This approach does not jeopardize stability while facilitating faster discovery of high-reward behaviors. Second, to prevent entropy collapse and allow rare but crucial reasoning/code tokens to grow, we replace symmetric clipping with an asymmetric interval $[1 - \varepsilon_{\text{low}}, 1 + \varepsilon_{\text{high}}]$ with a higher upper clipping threshold $\varepsilon_{\text{high}}$ than the lower threshold $\varepsilon_{\text{low}}$. This relaxes the trust region on probability increases while keeping a firm lower bound on decreases, which empirically improves executability in stage-1 RL training and supports generalization in stage-2 RL training. Following Specifically, we raise $\varepsilon_{\text{high}}$ to 0.28 during the training. Finally, to improve sample efficiency, critical for small-scale LLMs under limited data and verification budgets, instead of optimizing a sequence-averaged loss, we accumulate the loss over tokens and normalize by the total number of tokens in the group $\sum_i |o_i|$, which can yield denser learning signals for long outputs. The final loss of OptGRPO is as follows:

$$\mathcal{J}_{\text{OptGRPO}}(\theta) = \mathbb{E}_{q \sim P(Q), \{o_i\}_{i=1}^{G} \sim \pi_{\theta_{\text{old}}}(\cdot|q)} \frac{1}{\sum_{i=1}^{G} |o_i|} \sum_{i=1}^{G} \sum_{t=1}^{|o_i|}$$

$$\min \left[ \frac{\pi_\theta(o_{i,t}|q, o_{i,<t})}{\pi_{\theta_{\text{old}}}(o_{i,t}|q, o_{i,<t})} \hat{A}_{i,t}, \ \text{clip}\left(\frac{\pi_\theta(o_{i,t}|q, o_{i,<t})}{\pi_{\theta_{\text{old}}}(o_{i,t}|q, o_{i,<t})}, 1 - \varepsilon_{\text{low}}, 1 + \varepsilon_{\text{high}}\right) \hat{A}_{i,t} \right]. \tag{4}$$

## 4 EXPERIMENT

We evaluate MiniOpt models on diverse optimization benchmarks spanning multiple types and scenarios to assess whether small scale parameter LLMs (3B/7B/14B) can achieve strong optimization generalization ability. We report Solving Accuracy (SA) as primary metrics, SA presents the proportion of output solutions of executed code that are numerically equal to the optimal solution provided from the labels of benchmarks. We use Execution Rate (ER) as a supplement that shows the proportion of generated code samples that run successfully without errors. Comparisons cover general LLMs, general reasoning LLMs, prompting-based baselines and learning-based baselines. The experiments aim to answer four key questions below.

**(Q1) Optimization Generalization Ability at Small-Scale LLMs**: To what extent can MiniOpt at 3B/7B/14B achieve high SA across types and scenarios, and how does it compare with larger reasoning LLMs and prior learning-based approaches?

**(Q2) Pareto Front of Performance vs. Cost**: What is the limit of the scale of model parameters for achieving strong optimization generalization?

**(Q3) Importance of the Training Pipeline of MiniOpt**: How critical are the lightweight SFT warm-up and the two-stage RL to the performance of MiniOpt?

**(Q4) Importance of the Reasoning to Model and Solve Paradigm and OptReward**: How do the proposed paradigm and the corresponding Opt Reward contribute to boost SA and ER in solving optimization problems?

The four questions are answered sequentially in the following sections, which first provide a detailed introduction to the experiments and then present an analysis of the results.

### 4.1 EXPERIMENTAL SETUP

Since widely-used packages like Gurobi and COPT are close-source, generating code for them may entail licensing costs, and a model's generalization capability across different modeling languages and solvers depends on the proportion of relevant corpora in its pre-training and post-training data. Therefore, following established practices (Jiang et al., 2025), we adopted the open-source, solver-agnostic Pyomo modeling language as the language for our training and inference solver code.

As for the solvers employed, the key consideration lies in their ability to select the appropriate solver for a specific optimization type. For instance, GLPK only supports linear programming and integer programming and is incapable of solving nonlinear problems. Therefore, this paper selects four types of solvers to cover the solving requirements of as many optimization problem types as possible, so as to automatically adapt to the problem types when generating the solving code.

The evaluation encompasses nine benchmarks about operations research optimization: NL4Opt (Ramamonjison et al., 2022), Mamo (Easy and Complex subsets) (Huang et al., 2025b), IndustryOR (Huang et al., 2025a), NLP4LP (AhmadiTeshnizi et al., 2024), ComplexOR (Xiao et al., 2024), OptMATH (Lu et al., 2025), OptiBench (Yang et al., 2025b), and ICML Competition (Yang et al., 2024b). We follow the same setting in LLMOPT (Jiang et al., 2025) to ensure consistency and comparability. For the newly included datasets, OptMATH-Bench and OptiBench, we adhere to their original data splits provided by the authors. For ICML Competition dataset, we use it exclusively as out-of-distribution test data to evaluate the generalization capability of our model.

To validate the correctness of the solutions obtained by MiniOpt and all baselines, this paper compares their log files generated during the execution process against the ground-truth solutions provided by the benchmarks. A solution is deemed correct if the log file yields a non-empty match with the ground-truth; otherwise, it is considered unsuccessful.

### 4.2 ANALYSIS OF OPTIMIZATION GENERALIZATION

In this section, we compare MiniOpt with general LLMs (Qwen2.5 Series, DeepSeek-V3), general reasoning LLMs (Qwen3 Series, DeepSeek-R1, Gemini-2.5-Pro, GPT-5), prompt-based methods (Chain-of-Experts, OptiMUS, Reflexion), learning-based methods (OptMATH-7B, LLMOPT-14B, Step-OPt-8B/7B/3B), demonstrating the optimization generalization capability of MiniOpt. The

Table 1: Comparison of the SA metric across 9 benchmarks with rankings (NL4Opt, ICML Competition, Mamo Easy, Mamo Complex, NLP4LP, ComplexOR, IndustryOR, OptiBench, OptMATH-Bench). **Bold** indicates 1st, wavy underline indicates 2nd, underline indicates 3rd. "Rank*" represents the result of sorting methods among parameter sizes below 10B.

| Category | Models / Methods | Avg. | Rank | Rank* | NL4Opt | ICML.C | Mamo.E | Mamo.C | NLP4LP | Com.OR | Indus.OR | OptiBench | OptMATH-Bench |
|---|---|---|---|---|---|---|---|---|---|---|---|---|---|
| General Models | Qwen2.5-3B-Instruct | 9.98 | 22 | 11 | 19.13 | 18.78 | 17.18 | 2.37 | 18.60 | 0.00 | 2.00 | 11.74 | 0.00 |
| | Qwen2.5-7B-Instruct | 30.05 | 17 | 8 | 53.48 | 51.71 | 35.58 | 4.27 | 55.37 | 16.67 | 13.00 | 35.54 | 4.82 |
| | Qwen2.5-14B-Instruct | 43.59 | 12 | - | 67.39 | 63.17 | 80.21 | 15.64 | 67.36 | 22.22 | 22.00 | 41.65 | 12.65 |
| | DeepSeek-V3 | 57.81 | 4 | - | 78.26 | 77.56 | 84.82 | 26.54 | 79.34 | 44.44 | 26.00 | 64.13 | 39.16 |
| General Models (Thinking) | Qwen3-4B | 10.25 | 21 | 10 | 16.52 | 17.56 | 13.80 | 6.64 | 15.29 | 5.56 | 2.00 | 11.90 | 3.01 |
| | Qwen3-8B | 19.77 | 19 | 9 | 30.87 | 29.51 | 23.93 | 9.95 | 34.71 | 11.11 | 6.00 | 28.26 | 3.61 |
| | Qwen3-14B | 22.54 | 18 | - | 24.35 | 22.68 | 36.20 | 11.37 | 19.42 | 38.89 | 15.00 | 22.31 | 12.65 |
| | DeepSeek-R1 | 58.51 | 3 | - | 83.91 | 75.37 | 74.54 | 39.81 | 69.83 | 44.44 | 32.00 | 66.94 | 39.76 |
| | Gemini-2.5-Pro | 57.04 | 5 | - | 78.26 | 71.22 | 65.95 | 30.81 | 73.55 | 50.00 | 28.00 | 61.32 | 54.21 |
| | GPT-5 | 56.57 | 7 | - | 80.43 | 73.66 | 58.12 | 23.22 | 73.14 | 61.11 | 26.00 | 64.63 | 48.80 |
| Prompt-based Methods | Chain-of-Experts | 41.03 | 14 | - | 66.52 | 56.59 | 63.65 | 22.75 | 59.09 | 33.33 | 19.00 | 45.29 | 3.01 |
| | OptiMUS | 18.76 | 20 | - | 13.48 | 33.17 | 37.27 | 11.85 | 18.18 | 16.67 | 8.00 | 26.61 | 3.61 |
| | Reflexion | 41.28 | 13 | - | 56.52 | 52.20 | 84.82 | 18.01 | 53.72 | 38.89 | 19.00 | 41.16 | 7.23 |
| Learning-based Models | Step-OPT-LLaMA-3.2-3B | 39.61 | 15 | 6 | 70.00 | 26.59 | 68.10 | 36.49 | 63.64 | 38.89 | 17.00 | 26.78 | 9.04 |
| | Step-OPT-LLaMA-3-8B | 50.80 | 10 | 4 | 75.22 | 68.54 | 79.45 | 50.71 | 64.88 | 27.78 | 27.00 | 49.75 | 13.86 |
| | Step-OPT-Qwen2.5-3B | 36.15 | 16 | 7 | 41.30 | 38.54 | 75.31 | 20.85 | 48.35 | 27.78 | 21.00 | 40.00 | 7.23 |
| | Step-OPT-Qwen2.5-7B | 47.42 | 11 | 5 | 77.83 | 57.32 | 69.33 | 50.24 | 48.35 | 38.89 | 21.00 | 48.76 | 9.04 |
| | OptMATH-7B | 52.37 | 9 | 3 | 78.70 | 66.83 | 84.20 | 34.12 | 68.60 | 33.33 | 19.00 | 52.23 | 34.34 |
| | LLMOPT-14B | 54.81 | 8 | - | 80.28 | 75.35 | 89.53 | 44.08 | 73.42 | 35.29 | 29.00 | 53.83 | 12.50 |
| **Ours** | MiniOpt-3B | 56.94 | 6 | 2 | 83.04 | 68.05 | 85.43 | 35.07 | 73.55 | 50.00 | 21.00 | 53.55 | 42.77 |
| | MiniOpt-7B | 62.76 | 2 | 1 | 89.13 | 77.56 | 88.34 | 38.39 | 79.34 | 55.56 | 26.00 | 59.34 | 51.20 |
| | MiniOpt-14B | 66.10 | 1 | - | 92.17 | 86.34 | 90.80 | 33.65 | 79.75 | 61.11 | 27.00 | 67.44 | 56.63 |

information on these methods can be found in Appendix B. Tables 1 summarizes SA on nine benchmarks that span 7 optimization types and 22 scenarios, and the statistics on problem categories and scenarios are provided in Appendix A.2. The ER metrics for all the methods on the nine benchmarks and its analysis can be found in the Appendix F.

**Overall Performance (Answer to Q1).** Across all nine benchmarks, MiniOpt-7B achieved the strongest average performance among all baselines. Notably, MiniOpt-3B surpassed all prompt-based and learning-based methods. For instance, compared to LLMOPT-14B, the state-of-the-art learning-based model, MiniOpt-3B achieved an average SA that is 2.13% higher. When compared to DeepSeek-V3 and DeepSeek-R1, MiniOpt-3B scored 0.87% and 1.57% lower, respectively. Furthermore, as a higher-parameter variant of MiniOpt, MiniOpt-14B achieves the highest average SA of 66.10%, significantly raising the performance ceiling of MiniOpt in practical applications.

Besides, SIRL (Chen et al., 2025) is not included in Table 1 because the paper was accepted only shortly before our submission deadline. Although it proposes a reasoning model training pipeline, the important issue of improving the optimization generalization ability of small-scale models with limited data and computing resources has not been studied. According to the results presented in the SIRL paper (Chen et al., 2025), MiniOpt-7B outperforms the SIRL-Qwen2.5-7B model of similar size by 22.20% solving accuracy on the hard OptMATH-Bench benchmark.

**Competitiveness of Small-Scale Models (Answer to Q1).** MiniOpt remains competitive even at smaller scales. MiniOpt-3B reaches 56.94% SA on average, this performance already matching or exceeding several much larger reasoning models (e.g., the average SA higher than GPT-5 at 56.57%) and clearly outperforming general-purpose 14B post-trained models (e.g., +13.35% over Qwen2.5-14B-Instruct on average). Performance grows smoothly with scale under the same training pipeline, The average SA metric grows by +5.82% when parameters increase from 3B to 7B, and by +9.16% when further scaled to 14B.

**Challenging Benchmarks (Answer to Q1).** On the most demanding sets that require faithful modeling and solver usage, MiniOpt shows clear advantages. On the latest challenging benchmark OptMATH-Bench, MiniOpt-14B achieves 56.63% SA, outperforming Gemini-2.5-Pro (54.21%) and GPT-5 (48.80%). Even on extremely high-dimensional test sets such as Indus.OR, where small-parameter MiniOpt does not attain the highest SA due to limitations in instruction following, it still delivers competitive performance levels in both metrics.

**Breadth across Types and Scenarios (Answer to Q1).** To rigorously evaluate the optimization generalization, we analyze the SA of MiniOpt-7B and MiniOpt-3B across three optimization types and three application scenarios under two difficulty levels. At the medium-difficulty OptiBench,

MiniOpt-7B achieved 59% (M), 74% (T), 68% (L), and 65% (LP), 63% (IP), 45% in Mixed-Integer Linear Programming (MILP). MiniOpt-3B scored 55% (M), 71% (T), 59% (L), and 60% (LP), 58% (IP), 38% (MILP). On the hard-difficulty OptMATH-Bench, MiniOpt-7B achieved 58% (M), 56% (T), 57% (L), and 55% (LP), 100% (IP), 56% (MILP). MiniOpt-3B attained 43% (M), 56% (T), 76% (L), and 73% (LP), 67% (IP), 47% (MILP). These results reflect consistent and scalable generalization across optimization types, problem scenarios, and difficulty levels. In order to investigate whether the pipeline training in this paper would impair other abilities of the LLMs, we also discussed the generalization performance of MiniOpt-3B and 7B under different tasks except optimization problems. Detailed information is presented in Appendix J.

## 4.3 PARETO FRONT OF PERFORMANCE VS. COST

**Analysis of the Pareto Front (Answer to Q2).** Figure 2 and Figure 5 in Appendix H indicate that MiniOpt series (represented by the solid red line establishes a new and superior Pareto front in the performance-versus-cost trade-off. Since the parameter size of GPT-5 and Gemini-2.5-Pro have not been disclosed, we do not label these two models in the figures). As the scale of the model increases, the average SA performance of MiniOpt also grows steadily. The MiniOpt-14B variant achieves the highest average SA of 66.10% among all models. It achieves a comprehensive performance lead while having substantially fewer parameters than top-tier general reasoning models such as DeepSeek-R1. Compared to the similar modeling and solving model Step-OPT, Step-OPT-LLaMA-3-8B achieves an average SA of 50.80% across 9 benchmarks, while the proposed MiniOPT attains an average SA of 62.76%. When the parameter scale of both models are reduced to 3B, Step-OPT-LLaMA-3.2-3B exhibit a performance drop of 11.19%. For Step-OPT-Qwen2.5, the SA degradation is 11.27%, **while MiniOPT only decreased by 5.82%**. This indicates that the key advantage of MiniOPT lies in its ability to maintain superior performance even with reduced parameter scale. From the perspective of capability density (Xiao et al., 2025a), MiniOPT effectively achieves lower parameter requirements and inference costs while preserving comparable performance.

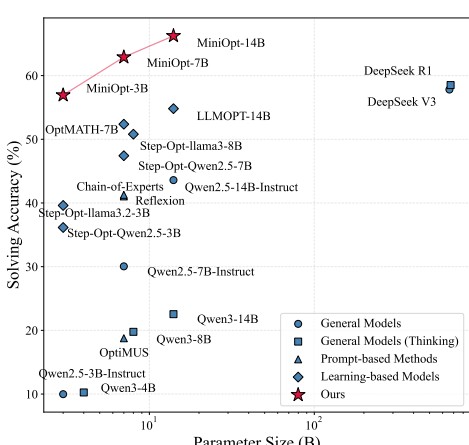

Solving Accuracy vs. Parameter Size

Figure 2: Comparison of average SA against model parameter scales for various methods. MiniOpt is the Pareto optimal among compared methods.

## 4.4 ABLATION STUDY

**Importance of the Reasoning to Model and Solve Paradigm and OptReward (Answer to Q4).** We ablate core components of MiniOpt-3B and report results of the SA metric in Table 2 and the ER metric in Table 6 in Appendix G. As evidenced in Table 2, each module of the proposed reasoning to model and solve paradigm demonstrates substantial contributions to modeling and solving optimization problems with smaller-scale models under limited training resources. Among these, RL provides the most significant improvement, highlighting the importance of the proposed paradigm. The reasoning to model and solve paradigm and OptReward together turn free-form generation into a verifiable formulation, which are easy to verify. Removing them and keeping only a final-answer signal (w/o OptReward) drops averages to 52.44% and 83.39% ($\Delta$ SA=–4.50%, $\Delta$ ER= –4.65%), respectively. The largest losses appear where the problems are challenging: Com.OR ($\Delta$ SA = –16.67%, $\Delta$ ER = –16.67%), Indus.OR ($\Delta$ SA = -3.00%, $\Delta$ ER = –6.00%), and OptMATH-Bench ($\Delta$ SA = –4.82%, $\Delta$ ER = –10.24%). These patterns align with the roles of the three reward components: the format score enforces the labels `<think>` and `<answer>` in the responses are complete; the five-element score shapes the intermediate blueprint, so the models learns to extract problem structure before coding; the Accuracy Score certifies numerical correctness by executing the Pyomo code and comparing the returned optimum with the reference. In combination, this reasoning to model and solve paradigm, together with verifiable reward, steers learning toward structurally consistent

Table 2: Ablation study (MiniOpt-3B) on the SA metric across 9 benchmarks.

| Category | Model / Method | Avg. | NL4Opt | ICML.C | Mamo.E | Mamo.C | NLP4LP | Com.OR | Indus.OR | OptiBench | OptMATH-Bench |
|---|---|---|---|---|---|---|---|---|---|---|---|
| Ablations | MiniOpt-3B | **56.94** | 83.04 | 68.05 | **85.43** | **35.07** | **73.55** | **50.00** | **21.00** | 53.55 | **42.77** |
| | MiniOpt-3B w/o SFT Warm-up | 53.59 | **83.91** | **70.00** | 83.28 | 33.65 | 68.18 | 38.89 | 15.00 | **53.88** | 35.54 |
| | MiniOpt-3B w/o RL | 39.25 | 52.61 | 48.29 | 80.06 | 25.12 | 60.33 | 16.67 | 15.00 | 34.71 | 20.48 |
| | MiniOpt-3B w/o Two-stage RL | 54.12 | 83.48 | 69.02 | 82.67 | 33.65 | 69.01 | 44.44 | 16.00 | **53.88** | 34.94 |
| | MiniOpt-3B w/o Data Selection | 53.50 | 78.70 | 68.05 | 84.82 | 30.81 | 72.31 | 38.89 | 19.00 | 52.89 | 36.75 |
| | MiniOpt-3B w/ Random Selection | 50.94 | 83.48 | 69.76 | 82.98 | 29.38 | 70.66 | 27.78 | 14.00 | 52.73 | 27.71 |
| | MiniOpt-3B w/o OptReward | 52.44 | 78.70 | 66.83 | 82.52 | 33.18 | 70.25 | 33.33 | 18.00 | 51.24 | 37.95 |
| | MiniOpt-3B w/ GRPO | 48.33 | 70.87 | 60.00 | 81.29 | 32.23 | 65.70 | 27.78 | 17.00 | 45.79 | 34.34 |
| | MiniOpt-3B w/ DAPO | 54.57 | 83.48 | 70.24 | 84.66 | 27.49 | 73.14 | 44.44 | 18.00 | 54.17 | 34.94 |

behaviors, yielding higher SA and ER across various optimization problems. For a detailed ablation analysis of the proposed model-and-solve paradigm, the training process, and the significance of OptReward and OptGRPO, please see Appendix G.

## 5 DISCUSSION

**Exploration of Models with Smaller Parameter Scale.** To further explore the performance of MiniOpt with smaller parameter scales, thereby better balancing the trade-off between parameters and performance, we deploy MiniOpt-1.5B with the same reasoning to model and solve paradigm (Section 3.2), OptReward (Section 3.4.1), and training pipeline (Section 3.3). The results shown in Table 7 and 8 in Appendix I demonstrate that when the model scale is reduced from 3B to 1.5B, MiniOpt achieves an average SA of 46.15% and an average ER of 80% across the 9 benchmarks, still surpassing all baselines except for learning-based methods, maintaining a strong competitiveness.

**Dimensionality Scalability Issue.** Our empirical results show that MiniOpt-3B can correctly model optimization problems with dimensionalities as high as 72 on the OptMATH-Bench task and 80 on the Mamo Complex task. This constitutes a remarkably strong performance for models of such limited scale, which highlights that the MiniOpt pipeline effectively enhances the model's optimization modeling capacity and retains robust generalization even in high-dimensional settings. For problems with even higher dimensionality, the descriptions of numerical parameters typically become substantially longer. Such extended input sequences inherently pose a significant challenge to the comprehension capabilities of small-scale models. As corroborating evidence, larger models such as MiniOpt-7B successfully solve problems up to 109 dimensions on the OptMATH-Bench task.

**Cost Savings of MiniOpt and Generality of Base Models.** We find that MiniOpt-3B achieves a 3.01% higher SA than DeepSeek-R1 on the OptMATH-Bench while using 62.4% fewer average output tokens. For MiniOpt-7B, the corresponding improvements are 11.44% higher SA and a 39.6% reduction in average output tokens. More details and discussions are provided in Appendix K. This demonstrates that MiniOpt constitutes a general framework capable of enabling small-scale models to achieve strong optimization generalization under constrained data and computational resources. Furthermore, although Qwen2.5 series models are adopted as the base model in this paper, we have also conducted experiments with different base models and obtained closely aligned results.

## 6 CONCLUSION

This paper proposes a novel reasoning to model and solve paradigm and the small scale model, MiniOpt-3B, achieves higher performance with a small-scale parameter and limited resources. We explore the optimization generalization of the model in various types of optimization, problem scenarios, and high-variable dimensions. Empirical results demonstrate that MiniOpt exhibits strong generalization performance under these varying conditions. Furthermore, this study explores the minimum parameter scale required for MiniOpt to maintain competitive performance. Future work includes exploring efficient modeling and solving methods for optimization problems with high-dimensional variables or a large number of constraints.

## ETHICS AND REPRODUCIBILITY STATEMENT

**Ethics**. This work does not involve any human subjects, personal data, or sensitive information. All the test datasets used are publicly available, and no proprietary or confidential information is used.

**Reproducibility**.   Experimental settings are described in Section 4.1 and datasets included in Appendix E. The code is available at `https://anonymous.4open.science/r/MiniOpt-6194`.

## LLM USAGE STATEMENT

No LLMs were used in the research ideation and paper writing of this work.

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

## A DATASETS

### A.1 THE INTRODUCTION OF EVALUATION DATASETS

In this section, we provide an overview of the datasets used for performance evaluation in our experiments. These datasets cover a wide range of optimization types and scenarios, ensuring the robustness and generalization of our proposed method. In our practice, we use the version of the benchmark datasets above from https://github.com/antgroup/LLMOPT.

**NL4Opt** (Ramamonjison et al., 2022) dataset is curated from the NL4Opt Competition. For this benchmark, we used the test split containing 230 annotated linear programming word problems

Table 3: Statistics of the optimization problem datasets

| Dataset Name | # of Data |
|---|---|
| NL4Opt | 230 |
| ICML.C | 410 |
| Mamo.E | 652 |
| Mamo.C | 211 |
| NLP4LP | 242 |
| Com.OR | 18 |
| Indus.OR | 100 |
| OptiBench | 605 |
| OptMATH-Bench | 166 |

after manually removing 15 unsolvable problems from the original 245 problems. Each problem is sourced from domains such as sales, advertising, and investment, ensuring a balanced representation.

**Mamo** (Huang et al., 2025b) dataset (optimization split of the original Mamo dataset) consists of two parts: Easy_LP and Complex_LP. These two subsets provide 652 high-school-level and 211 undergraduate-level Linear Programming (LP) and Mixed-Integer Linear Programming (MILP) problems, respectively.

**IndustryOR** (Huang et al., 2025a) is the first industrial dataset specifically designed for optimization modeling. It incorporates data from 13 different industries and covers a variety of real-world scenarios. The dataset includes real operations research problems from eight different industries, covering five types of optimization problems, and divided into three difficulty levels. The test dataset contains 100 instances with optimal solutions.

**NLP4LP** (AhmadiTeshnizi et al., 2024) dataset includes 242 feasible samples sourced from optimization textbooks and lecture notes. These problems cover areas such as facility location, network flow, scheduling, and portfolio management. Each instance in NLP4LP includes a description, sample parameter data file, and optimal value derived from textbook solutions or manual solving, offering a range of complex optimization challenges of varying difficulty levels.

**ComplexOR** (Xiao et al., 2024) dataset is developed in collaboration with three specialists in operations research. It contains 18 samples sourced from diverse references such as academic papers, textbooks, and real-world industrial scenarios. These problems encompass a broad spectrum of topics, including supply chain optimization, scheduling problems, and warehousing optimization, providing comprehensive and complex optimization challenges.

**OptMATH-Bench** (Lu et al., 2025) is a large-scale, challenging benchmark specifically designed to evaluate the optimization modeling capabilities of LLMs, encompassing diverse optimization problem types across 10+ real-world application domains such as logistics, manufacturing, transportation, and finance. The benchmark features significantly more complex problem descriptions with an average length 2.9× longer than Mamo Easy, containing extended natural language contexts and intricate constraints that pose greater challenges.

**OptiBench** (Yang et al., 2025b) is a comprehensive benchmark for evaluating large language models' end-to-end optimization problem-solving capabilities. The dataset contains 605 carefully curated optimization problems that span multiple optimization types and formats. OptiBench includes problems of Linear Programming (LP), Integer Programming (IP), and Mixed-Integer Linear Programming (MILP), encompassing a wide range of optimization complexities.

**ICML Competition** (Yang et al., 2024b) dataset comprises data from the ICML 2024 Challenges on Automated Math Reasoning - Track 3: Automated Optimization curated from the competition's test split. Since the original ground truth is not released by the organizers, all solutions in this dataset are manually labeled. The dataset serves as a challenging benchmark for evaluating end-to-end optimization reasoning and problem-solving capabilities of language models.

### A.2 THE DISTRIBUTION OF OPTIMIZATION TYPES AND PROBLEM SCENARIOS OF BENCHMARKS

To evaluate the generalization ability of the MiniOpt across different problem scenarios through experiments, this paper has counted the number of optimization types and scenarios in 9 benchmarks.

The distribution histogram of optimization types in the nine benchmarks used in this paper is shown in Figure 3, and the distribution histogram of problem scenarios is shown in Figure 4.

| Type | Com.OR | ICML.C | Indus.OR | Mamo.C | Mamo.E | NL4Opt | NLP4LP | OptMAT H-Bench | Optibench |
|------|--------|--------|----------|--------|--------|--------|--------|----------------|-----------|
| CO | 4 | 8 | 12 | 39 | 0 | 0 | 0 | 19 | 11 |
| IP | 4 | 211 | 23 | 11 | 2 | 161 | 167 | 7 | 291 |
| LP | 9 | 175 | 31 | 123 | 650 | 69 | 75 | 16 | 214 |
| MILP | 1 | 16 | 30 | 38 | 0 | 0 | 0 | 91 | 44 |
| MOP | 0 | 0 | 4 | 0 | 0 | 0 | 0 | 0 | 0 |
| NLP | 0 | 0 | 0 | 0 | 0 | 0 | 0 | 0 | 45 |
| SOCP | 0 | 0 | 0 | 0 | 0 | 0 | 0 | 33 | 0 |
| # Data | 18 | 410 | 100 | 211 | 652 | 230 | 242 | 166 | 605 |

Figure 3: Histogram showing the distribution of optimization types across 9 benchmarks. We categorize the problems in the benchmarks into these types: Combinatorial Optimization (CO), Integer Programming (IP), Linear Programming (LP), Mixed-Integer Linear Programming (MILP), Multi-Objective Optimization Problems (MOP), Nonlinear Programming (NLP), Second-Order Cone Programming (SOCP).

| Scenario | Com.OR | ICML.C | Indus.OR | Mamo.C | Mamo.E | NL4Opt | NLP4LP | OptMAT H-Bench | Optibench |
|----------|--------|--------|----------|--------|--------|--------|--------|----------------|-----------|
| Agriculture | 0 | 37 | 5 | 4 | 31 | 13 | 12 | 0 | 46 |
| Aviation | 3 | 2 | 2 | 2 | 0 | 2 | 1 | 12 | 2 |
| Construction | 0 | 1 | 1 | 1 | 42 | 1 | 1 | 0 | 8 |
| Education | 0 | 5 | 1 | 0 | 30 | 0 | 0 | 0 | 7 |
| Energy | 0 | 0 | 0 | 5 | 25 | 1 | 2 | 7 | 15 |
| Environment | 0 | 0 | 0 | 0 | 31 | 0 | 1 | 0 | 0 |
| Finance | 0 | 16 | 7 | 5 | 85 | 5 | 5 | 6 | 22 |
| Healthcare | 0 | 15 | 1 | 5 | 31 | 23 | 29 | 0 | 20 |
| Logistics | 1 | 27 | 8 | 32 | 25 | 21 | 23 | 22 | 42 |
| Manufacturing | 5 | 169 | 37 | 12 | 34 | 77 | 83 | 56 | 254 |
| Marketing | 1 | 9 | 0 | 0 | 43 | 2 | 2 | 0 | 11 |
| Military | 0 | 0 | 2 | 0 | 36 | 0 | 0 | 0 | 3 |
| Public Utilities | 0 | 9 | 1 | 4 | 7 | 2 | 2 | 2 | 19 |
| Resources | 0 | 15 | 0 | 12 | 9 | 11 | 8 | 3 | 17 |
| Retail | 0 | 16 | 4 | 2 | 31 | 8 | 9 | 1 | 19 |
| Services | 4 | 38 | 10 | 6 | 73 | 29 | 29 | 7 | 40 |
| Sports | 0 | 0 | 0 | 0 | 31 | 0 | 0 | 0 | 0 |
| Supply Chain | 2 | 12 | 7 | 48 | 31 | 5 | 3 | 11 | 11 |
| Technology | 0 | 0 | 1 | 3 | 0 | 0 | 0 | 1 | 3 |
| Telecommunications | 1 | 1 | 0 | 8 | 27 | 0 | 0 | 9 | 1 |
| Transportation | 1 | 31 | 8 | 57 | 45 | 23 | 26 | 9 | 40 |
| Other | 0 | 7 | 5 | 5 | 15 | 7 | 6 | 20 | 25 |
| # Data | 18 | 410 | 100 | 211 | 682 | 230 | 242 | 166 | 605 |

Figure 4: Histogram showing the distribution of optimization problem scenarios across 9 benchmarks. We categorize the problems in the benchmarks into these scenarios: Agriculture, Aviation, Construction, Education, Energy, Environment, Finance, Healthcare, Logistics, Manufacturing, Marketing, Military, Public Utilities, Resources, Retail, Services, Sports, Supply Chain, Technology, Telecommunications, Transportation, Other.

### A.3 TRAINING DATASETS FOR SFT WARM-UP AND TWO-STAGE RL

The data volume and data sources for each stage of MiniOpt's training process are illustrated in Table 4.

Table 4: The number of samples in the dataset for training. Note that all data mentioned in the table comes entirely from the training split of the corresponding dataset.

| Training Stage | Dataset Size | Data Source |
|----------------|--------------|-------------|
| SFT Warm-up | 140K | OptMATH-Train |
| RL-Stage 1 | 1585 | NL4Opt (Train) & ICML.C (Train) |
| RL-Stage 2 | 3000 | OptMATH-Train |

## B    BASELINES

### B.1    GENERAL MODELS

**Qwen2.5-3B/7B/14B** (Yang et al., 2024a). The models of the Qwen2.5 series are widely adopted as base models or baselines. The series showcases significant enhancements such as substantially improved knowledge, coding, and mathematical capabilities. Key features excel at instruction following, processing long contexts up to 128K tokens, and robustly handling structured data like JSON.

**DeepSeek-V3** (DeepSeek-AI et al., 2024). DeepSeek-V3 introduces a sparse Mixture-of-Experts (MoE) model with 671B total parameters. It achieves training efficiency through Multi-head Latent Attention (MLA) architecture and an auxiliary-loss-free load balancing strategy. Pretrained on 14.8T tokens with an SFT and reinforcement learning (RL).

### B.2    GENERAL REASONING MODELS

**Qwen3-4B/8B/14B** (Yang et al., 2025a). Qwen3 pioneers a unified architecture (0.6B to 235B) integrating thinking mode (complex reasoning) and non-thinking mode (rapid responses) with dynamic switching. Its thinking budget mechanism enables adaptive computational allocation. The series outperforms larger MoE models in tasks such as coding, mathematics, and agent application.

**DeepSeek-R1** (DeepSeek-AI et al., 2025).The DeepSeek-R1 is an enhanced model based on DeepSeek-R1-Zero presented in this work. It's a purely RL-driven reasoning model requiring no SFT pretraining. As the most representative model with thinking ability, DeepSeek-R1 is an important baseline for reasoning models.

**Gemini-2.5-Pro** (Comanici et al., 2025). Gemini-2.5-Pro is a powerful multimodal agent that has excellent programming / reasoning performance and enables the processing of long video content. The Gemini-2.5 family spans the full Pareto front of capability-cost optimization. Its integration of long-context understanding, multimodality, and reasoning unlocks novel agentic applications.

**GPT-5** (OpenAI, 2025). GPT-5 is the latest unified, router-mediated system that instantiates a spectrum of language-model instances ranging from a high-throughput, low-latency model (gpt-5-main) to a deliberative, compute-intensive reasoning model (gpt-5-thinking). The router selects the appropriate instantiation by conditioning on conversation type, task complexity, tool requirements, and explicit user directives, thereby optimizing both instruction adherence and inference efficiency.

### B.3    PROMPT-BASED METHODS

**Reflection** (Shinn et al., 2023). Reflexion is an enhanced language agent framework utilizing feedback mechanisms. It enables agents to excel at sequential decision-making tasks through task feedback analysis and memory buffering without requiring weight updates. This framework accommodates diverse feedback signals and demonstrates effectiveness across programming, math problems and language reasoning domains.

**OptiMUS** (AhmadiTeshnizi et al., 2024). OptiMUS is a highly modular solver that leverages the text understanding and generating capabilities of LLMs. It constructs specialized agents for entity extraction, mathematical modeling, and code generation using concise prompts, while incorporating a reflection mechanism for iterative improvement.

**Chain-of-Experts** (Xiao et al., 2024). Chain-of-Experts is a multi-agent framework specifically designed for operations research optimization problems. The system features a central controller that coordinates an interaction sequence among specialized agents, including a term interpreter, modeling agent, and programming expert. Thus solving optimization problems through precise coordination of multiple modules.

### B.4    LEARNING-BASED MODELS

**LLMOPT-14B** (Jiang et al., 2025). LLMOPT is a novel framework for optimization problem solving that leverages LLMs. It begins by formulating a unified representation of optimization prob-

lems, thereby enhancing the model's ability to generalize across diverse types of scenarios. Based on this unified description of five-element formulation, the framework generates the solving code. LLMOPT uses multi-instruction SFT and KTO alignment during training to enhance modeling accuracy and reduce model hallucinations.

**OptMATH-Qwen2.5-7B** (Lu et al., 2025). OptMATH-Qwen2.5-7B is trained end-to-end on the OptMATH-Train dataset, it generates both mathematical formulations and solver code from problem descriptions. The input consists of textual problem specifications, while the target output comprises concatenated sequences. Optimization follows the standard sequence-to-sequence loss function, enabling single-stage joint optimization of formulation and code generation.

**Step-OPT-8B** (Wu et al., 2025). Step-OPT is a model trained on a meticulously curated high-quality dataset. The training set is enhanced through Scope-Evolve and Complexity-Evolve techniques, which improve both the difficulty level and the coverage of application scenarios. Moreover, a multi-agent and stepwise verification mechanism is employed to enhance the quality of problems and solutions while eliminating errors present in the original dataset. Finally, supervised fine-tuning was performed on this high-quality training dataset resulting in the Step-OPT-8B model.

## C   THE DETAILS OF THE FIVE-ELEMENT FORMULATION

The five-element modeling formulation is a universal mathematical model for optimization problems, which consists of five parts. We start from the following formulation:

$$\min_{\boldsymbol{x} \in \mathcal{X} \subseteq \mathbb{R}^D} f(\boldsymbol{x}), \quad \text{s.t.} \ G(\boldsymbol{x}) \leq \boldsymbol{c}, \tag{5}$$

where $\boldsymbol{x} = (x_1, x_2, \ldots, x_D)^\top$ is the $D$-dimensional decision variable, $\mathcal{X}$ is the feasible region, $f : \mathcal{X} \to \mathbb{R}$ is the objective, $G(\boldsymbol{x}) : \mathbb{R}^D \to \mathbb{R}^m$ collects the constraints, and $\boldsymbol{c} \in \mathbb{R}^m$ provides the upper bounds. Among them, **Variables**, **Objective**, and **Constraints** correspond to $\boldsymbol{x}$, $f(\boldsymbol{x})$, and $G(\boldsymbol{x})$, while **Sets** and **Parameters** provide indices and numerical tables that instantiate and vectorize $f$ and $G$. Sets determine the dimensions and naming of decision and constraint families. Parameters supply exogenous constants such as costs, coefficients, budgets, and demands. Variables specify domains and bounds(e.g., continuous, nonnegative, integer, or binary), which jointly define the feasible region $\mathcal{X}$; domain-type restrictions such as "positive integers" may equivalently be encoded as explicit constraints, and our parser maps both styles to $\mathcal{X}$. Objective gives the minimization or maximization expression, and Constraints provide named families of linear or nonlinear equalities/inequalities composing $G$ and the bound vector $\boldsymbol{c}$. This representation naturally spans LP/IP/MILP: integrality arises through $\mathcal{X}$, linearity or nonlinearity is captured by the form of $G$, and multi-objective problems can be accommodated by extending $f(x)$ to a vector $F(x)$ with a scalarization scheme. The think output ends with the five-element formulation, which serves as a modeling blueprint.

## D   DETAILS OF GRPO ALGORITHM

This paper proposes OptGRPO based on the improvement of the GRPO algorithm (Shao et al., 2024). For each query, GRPO sample $G$ responses, compute the mean and standard deviation of their scalar rewards, and form a group-normalized advantage $\hat{A}_i = \frac{(r_i - \mu)}{\sigma}$. Specifically, GRPO optimizes the policy $\pi_\theta$ as follows:

$$\mathcal{J}_{\text{GRPO}}(\theta) = \mathbb{E}_{q \sim P(Q), \{o_i\}_{i=1}^G \sim \pi_{\theta_{\text{old}}}(\cdot | q)}$$

$$\left[ \sum_{i=1}^G \sum_{t=1}^{|o_i|} \frac{1}{|o_i|} \left( \min \left[ \frac{\pi_\theta(o_{i,t}|q, o_{i,<t})}{\pi_{\theta_{\text{old}}}(o_{i,t}|q, o_{i,<t})} \hat{A}_{i,t}, \ \text{clip}(\frac{\pi_\theta(o_{i,t}|q, o_{i,<t})}{\pi_{\theta_{\text{old}}}(o_{i,t}|q, o_{i,<t})}, 1 - \varepsilon, 1 + \varepsilon) \hat{A}_{i,t} \right] \right. \right.$$

$$\left. \left. - \beta D_{\text{KL}}[\pi_\theta(\cdot|q) \| \pi_{\text{ref}}(\cdot|q)] \right) \right], \tag{6}$$

where $q$ denotes queries sampled from the input dataset and $o$ denotes the model's outputs. $\varepsilon$ is the clipping threshold, $\beta$ is the coefficient on the KL penalty, and $D_{\mathrm{KL}}$ is the KL divergence between the current policy $\pi_\theta$ and the reference policy $\pi_{\mathrm{ref}}$.

## E   THE PROCESSING PIPELINE OF TRAINING DATA

The processing pipeline for the SFT training set is detailed in Section 3.3.1. The resulting collection of this pipeline is named OptMATH-Train-Pyomo, which contains approximately 140K samples. The prompt template used in code conversion is introduced in Appendix L, and the prompt of the solver adapter is introduced in Appendix M.

In the first stage of MiniOpt's RL training, we employ a set of relatively easy and small-scale optimization problems, for which the accuracy score is more readily maximized. This, in turn, incentivizes the model to attain higher format score and five-element score, thereby accelerating mastery of the reasoning to model and solve paradigm. Concretely, the dataset of the first stage is the union of the NL4Opt (AhmadiTeshnizi et al., 2024) and ICML Competition (Yang et al., 2024b) training splits, comprising 1585 problems. Each instance is presented as a natural-language prompt paired with a reference answer and is fully compatible with our pipeline of parsing, executing, and scoring, enabling straightforward computation of the OptReward.

The second stage of RL targets optimization generalization (Jiang et al., 2025) under a limited training budget. The objective is to construct a training set that simultaneously covers diverse optimization types and scenarios, while preserving the scenario proportions observed in the real distribution. Starting from the OptMATH-Train pool containing 201K problems, we label each instance with types and scenarios using the DeepSeek-V3 (DeepSeek-AI et al., 2024), with prompt templates and data distributions provided in Appendix O. We then sample a 3000-instance stage-2 training set subject to two constraints: (i) type-uniform coverage, with exactly 600 instances per type; and (ii) within each type, the scenario frequencies match the distribution in the full pool. Formally, letting $p(s)$ denote the overall scenario distribution in the complete pool, the target number of samples for each scenario $s$ under type $t$ is allocated as $n_t(s) = \mathrm{round}\big(600 \cdot p(s)\big)$, where $\sum_s n_t(s) = 600$.

To prevent leakage, problems used during the SFT warm-up (Section 3.3.1) are excluded from consideration. The resulting 3K training set is uniform across types and aligned with the scenario distribution of the data pool. Compared with random or non-selected baselines, this selection improves overall performance of MiniOpt under the same compute budget.

## F   COMPARISON OF EXECUTION RATE ACROSS 9 BENCHMARKS

This section we use Execution Rate (ER), the proportion of generated code samples that run successfully without errors. As shown in the Table 5, MiniOpt also exhibits superior text generation capabilities compared to baseline methods, which suggests its excellent code generation performance given the problem modeling.

## G   ABLATION STUDY OF MINIOPT ACROSS 9 BENCHMARKS

**Importance of the Training Pipeline of MiniOpt (Answer to Q3).** In the training pipeline of MiniOpt, each module plays a distinct role in improving SA and ER. First, the lightweight SFT warm-up provides a better starting point for RL training. Without it, averages for SA and ER fall to 53.59% and 83.66%, respectively. The decreases are $\Delta$SA=–3.35 and $\Delta$ER=–4.38. Second, removing all RL training collapses the average performance from SA of 56.94% and ER of 88.04% to 39.25% and 75.36%, respectively. With the sharpest drops on challenging datasets like Com.OR ($\Delta$SA = –33.33%, $\Delta$ER = –33.33%) and OptMATH-Bench ($\Delta$SA = –22.29%, $\Delta$ER = –34.33%) where accurate modeling and solver selection are indispensable. Third, collapsing the two-stage RL removes the progressive training that first consolidates the paradigm (stage-1) and then targets generalization (stage-2), averages drop to the SA of 54.12% and ER of 84.38% ($\Delta$ SA=–2.82%, $\Delta$ ER=–3.66%). Moreover, the data selection of stage-2 is crucial for sample efficiency that training on the full pool (w/o Data Selection) yield the average SA of 53.58% and the average ER of 82.04% ($\Delta$SA = –3.36%, $\Delta$ER = –6.00%), while the random selected training data (w/ Random

Table 5: Comparison of the ER metric across 9 benchmarks with rankings (NL4Opt, ICML Competition, Mamo Easy, Mamo Complex, NLP4LP, ComplexOR, IndustryOR, OptiBench, OptMATH-Bench). **Bold** indicates 1st, ~~wavy underline~~ indicates 2nd, underline indicates 3rd. "Rank*" represents the result of sorting methods among parameter sizes below 10B.

| Category | Model / Method | Avg. | Rank | Rank* | NL4Opt | ICML.C | Mamo.E | Mamo.C | NLP4LP | Com.OR | Indus.OR | OptiBench | OptMATH-Bench |
|---|---|---|---|---|---|---|---|---|---|---|---|---|---|
| General Models | Qwen2.5-3B-Instruct | 17.11 | 21 | 10 | 31.30 | 28.54 | 21.47 | 7.11 | 28.93 | 5.56 | 10.00 | 20.50 | 0.60 |
| | Qwen2.5-7B-Instruct | 41.69 | 18 | 8 | 66.09 | 68.54 | 40.34 | 19.43 | 71.90 | 16.67 | 30.00 | 53.22 | 9.04 |
| | Qwen2.5-14B-Instruct | 63.53 | 14 | - | 82.17 | 80.73 | 86.66 | 59.24 | 84.71 | 38.89 | 52.00 | 61.49 | 25.90 |
| | DeepSeek-V3 | 83.50 | 7 | - | 97.83 | 97.32 | 96.63 | 71.56 | 97.93 | 72.22 | 70.00 | 84.79 | 63.25 |
| General Models (Thinking) | Qwen3-4B | 14.15 | 22 | 11 | 19.57 | 20.00 | 15.18 | 18.48 | 18.18 | 11.11 | 5.00 | 16.20 | 3.61 |
| | Qwen3-8B | 25.55 | 20 | 9 | 36.96 | 38.29 | 26.53 | 18.01 | 39.26 | 22.22 | 10.00 | 35.04 | 3.61 |
| | Qwen3-14B | 31.17 | 19 | - | 28.70 | 25.12 | 40.18 | 30.81 | 22.73 | 61.11 | 27.00 | 27.44 | 17.47 |
| | DeepSeek-R1 | 83.07 | 9 | - | 96.09 | 94.15 | 89.72 | 84.83 | 91.32 | 61.11 | 82.00 | 88.92 | 53.01 |
| | Gemini-2.5-Pro | 89.65 | 4 | - | 94.35 | 96.10 | 95.86 | 86.26 | 96.28 | 77.78 | **87.00** | ~~91.90~~ | ~~81.32~~ |
| | GPT-5 | 83.26 | 8 | - | 98.08 | 97.32 | 71.47 | 55.92 | 97.52 | **88.89** | 78.00 | **92.07** | 69.88 |
| Prompt-based Methods | Chain-of-Experts | 61.72 | 15 | - | 79.57 | 73.41 | 72.85 | 56.87 | 77.27 | 55.56 | 54.00 | 65.45 | 20.48 |
| | OptiMUS | 52.13 | 17 | - | 45.22 | 75.85 | 73.77 | 43.60 | 44.63 | 44.44 | 48.00 | 65.95 | 27.71 |
| | Reflexion | 80.42 | 10 | - | 91.74 | 91.46 | 97.55 | 67.77 | 95.45 | 83.33 | 66.00 | 81.65 | 48.80 |
| Learning-based Models | Step-OPT-LLaMA3.2-3B | 67.44 | 13 | 6 | 98.70 | 48.05 | 90.64 | 73.93 | 99.17 | 66.67 | 60.00 | 48.76 | 21.08 |
| | Step-OPT-LLaMA-3-8B | 78.29 | 11 | 4 | 96.96 | 93.90 | **98.93** | 90.05 | 94.63 | 61.11 | 67.00 | 73.72 | 28.31 |
| | Step-OPT-Qwen2.5-3B | 59.37 | 16 | 7 | 54.78 | 58.78 | 97.09 | 42.18 | 81.40 | 61.11 | 50.00 | 67.27 | 21.69 |
| | Step-OPT-Qwen2.5-7B | 71.76 | 12 | 5 | 94.78 | 77.56 | 87.73 | 84.83 | 68.18 | 72.22 | 64.00 | 71.24 | 25.30 |
| | OptMATH-7B | 85.07 | 6 | 3 | 99.13 | 95.85 | 98.47 | 90.05 | 99.17 | 66.67 | 69.00 | 82.81 | 64.46 |
| | LLMOPT-14B | 90.03 | 3 | - | 97.42 | 93.90 | 92.29 | 77.73 | 97.93 | **88.89** | 61.00 | 73.22 | 31.93 |
| **Ours** | MiniOpt-3B | 88.04 | 5 | ~~2~~ | **99.57** | 95.85 | 98.47 | 87.68 | 99.59 | **88.89** | 70.00 | 83.64 | 68.67 |
| | MiniOpt-7B | ~~90.61~~ | 2 | **1** | **99.57** | ~~98.05~~ | **98.93** | ~~95.26~~ | **100.00** | **88.89** | 74.00 | 84.30 | 76.51 |
| | MiniOpt-14B | **92.35** | **1** | - | **99.57** | **98.54** | **98.93** | **97.16** | **100.00** | **88.89** | 77.00 | 89.09 | **81.93** |

Table 6: Ablation study (MiniOpt-3B) on the ER metric across 9 benchmarks.

| Category | Model / Method | Avg. | NL4Opt | ICML.C | Mamo.E | Mamo.C | NLP4LP | Com.OR | Indus.OR | OptiBench | OptMATH-Bench |
|---|---|---|---|---|---|---|---|---|---|---|---|
| Ablations | MiniOpt-3B | 88.04 | **99.57** | 95.85 | 98.47 | **87.68** | 99.59 | **88.89** | 70.00 | 83.64 | **68.67** |
| | MiniOpt-3B w/o SFT Warm-up | 83.66 | 99.13 | **96.59** | 98.16 | 84.36 | 98.35 | 66.67 | 68.00 | 84.46 | 57.23 |
| | MiniOpt-3B w/o RL | 75.36 | 96.09 | 92.68 | 97.09 | 69.19 | 97.93 | 55.56 | 62.00 | 73.39 | 34.34 |
| | MiniOpt-3B w/o Two-stage RL | 84.38 | 98.26 | 96.10 | 97.85 | 84.36 | 98.35 | 72.22 | 70.00 | 84.46 | 57.83 |
| | MiniOpt-3B w/o Data Selection | 82.04 | **99.57** | 95.61 | 96.78 | 86.73 | 98.76 | 55.56 | 66.00 | 82.15 | 57.23 |
| | MiniOpt-3B w/ Random Selection | 82.01 | 99.13 | 96.10 | 97.09 | 77.73 | 97.93 | 67.67 | 69.00 | 84.63 | 48.80 |
| | MiniOpt-3B w/o OptReward | 83.39 | 98.26 | 95.61 | 97.70 | 83.89 | 97.93 | 72.22 | 64.00 | 82.48 | 58.43 |
| | MiniOpt-3B w/ GRPO | 80.00 | 97.39 | 94.63 | 96.32 | 83.41 | 95.87 | 55.56 | 64.00 | 81.65 | 51.20 |
| | MiniOPT-3b w/ DAPO | 86.82 | 99.13 | 96.34 | **98.77** | 85.31 | 99.17 | 83.33 | **74.00** | **86.94** | 58.43 |

Selection) yield SA of 50.94% and ER of 82.01% on average (ΔSA = -6.00%, ΔER = -6.03%), indicating that type-uniform, globally scenario-aligned sampling concentrates updates where they best improve cross-type, cross-scenario behavior. Finally, reverting our OptGRPO to the original GRPO (w/o OptGRPO) further decreases both metrics, averages SA of 48.33% and ER of 80.00%. The drop aligns with our algorithmic choices: removing KL part frees exploration for small-scale LLMs; Clip-Higher part prevents entropy collapse by allowing probability increases on rare but crucial reasoning/code tokens; Token-Level Loss part can enhance the impact of long output, which is conducive to the training of the reasoning model. Together these changes improve sample efficiency and training stability, which is critical for eliciting strong optimization generalization at small parameter scales.

# H   COMPARISON OF AVERAGE ER AGAINST MODEL PARAMETER SCALES FOR VARIOUS METHODS

In this chapter, we present a comparative plot of the model parameters scale and the average ER across multiple methods, as illustrated in Figure 5.

# I   COMPARISON OF RESULTS BETWEEN THE MINIMUM MODEL AND MODELS OF OTHER SCALES

As shown in Tables 7 and 8, we compared the SA and ER of four different sizes of MiniOpt (1.5B, 3B, 7B, and 14B) across 9 benchmarks, respectively.

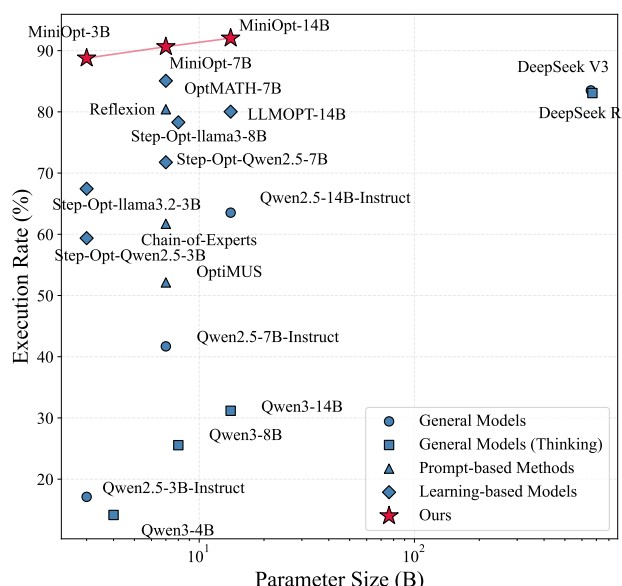

Execution Rate vs. Parameter Size

Figure 5: Comparison of average ER against model parameter scales for various methods. MiniOpt is the Pareto optimal among compared methods.

Table 7: Comparison of the SA metric between MiniOpt-1.5B and larger scale counterparts across 9 benchmarks.

| Solving Accuracy (SA) | | Avg. | NL4Opt | ICML.C | Mamo.E | Mamo.C | NLP4LP | Com.OR | Indus.OR | OptiBench | OptMATH-Bench |
|---|---|---|---|---|---|---|---|---|---|---|---|
| Dataset size | | | 230 | 410 | 652 | 211 | 242 | 18 | 100 | 605 | 166 |
| Ours | MiniOpt-1.5B | 46.15 | 63.48 | 55.37 | 77.15 | 27.01 | 58.68 | 38.89 | 16.00 | 41.98 | 36.75 |
| | MiniOpt-3B | 56.94 | 83.04 | 68.05 | 85.43 | 35.07 | 73.55 | 50.00 | 21.00 | 53.55 | 42.77 |
| | MiniOpt-7B | 62.76 | 89.13 | 77.56 | 88.34 | 38.89 | 79.34 | 55.56 | 26.00 | 59.34 | 51.20 |
| | MiniOpt-14B | 66.10 | 92.17 | 86.34 | 90.80 | 33.65 | 79.75 | 61.11 | 27.00 | 67.44 | 56.63 |

## J   THE SEESAW ISSUE OF LLMS

We assess whether adapting our models to optimization modeling introduces a seesaw issue for small-parameter models. After evaluating pre-training and post-training scores on six widely used general-purpose benchmarks: MMLU (Hendrycks et al., 2021a), MATH (Hendrycks et al., 2021b), HumanEval (Chen et al., 2021), TriviaQA (Joshi et al., 2017), RACE (Lai et al., 2017), GSM8K (Cobbe et al., 2021), we observe that the average scores of MiniOpt-7B and MiniOpt-3B decrease by only 1.83% and 3.05%, respectively. Notably, the 3B model even exhibits a 0.7% improvement on GSM8K, a proxy for mathematical reasoning. These results indicate that our training paradigm for optimization modeling does not induce a pronounced seesaw effect. We attribute this to two factors. First, the two-stage RL framework coupled with a verifiable OptReward constrains learning to structural modeling correctness and executable solving efficacy, mitigating overfitting to superficial linguistic style or lengthy chain-of-thought and thereby substantially reducing the risks of catastrophic forgetting and cross-task seesaw effects. Second, the foundational competencies required for optimization modeling, such as mathematical understanding, symbolic reasoning, program synthesis, and execution, which highly overlap with those assessed by general benchmarks such as MMLU, MATH, GSM8K, and HumanEval. Consequently, targeted reinforcement in this domain does not overwrite existing representations, instead, it yields small positive transfer on tasks closely aligned with modeling and solving, as exemplified by GSM8K.

Table 8: Comparison of the ER metric between MiniOpt-1.5B and larger scale counterparts across 9 benchmarks.

| Execution Rate (ER) | Avg. | NL4Opt | ICML.C | Mamo.E | Mamo.C | NLP4LP | Com.OR | Indus.OR | OptiBench | OptMATH-Bench |
|---|---|---|---|---|---|---|---|---|---|---|
| Dataset size | | 230 | 410 | 652 | 211 | 242 | 18 | 100 | 605 | 166 |
| Ours MiniOpt-1.5B | 80.00 | 93.91 | 91.71 | 97.09 | 79.15 | 91.74 | 66.67 | 63.00 | 75.87 | 60.84 |
| MiniOpt-3B | 88.04 | 99.57 | 95.85 | 98.47 | 87.68 | 99.59 | 88.89 | 70.00 | 83.64 | 68.67 |
| MiniOpt-7B | 90.61 | 99.57 | 98.05 | 98.93 | 95.26 | 100.00 | 88.89 | 74.00 | 84.30 | 76.51 |
| MiniOpt-14B | 92.35 | 99.57 | 98.54 | 98.93 | 97.16 | 100.00 | 88.89 | 77.00 | 89.09 | 81.93 |

## K    DETAILED DISCUSSION ON THE EFFICIENCY OF MINIOPT TO MODEL AND SOLVE OPTIMIZATION PROBLEMS

To validate the high efficiency of the MiniOpt model during inference, we compared the average number of tokens generated in full responses on the OptMATH dataset by two small-parameter models of MiniOpt and DeepSeek-R1. As shown in Table 9, MiniOpt achieves higher ER and SA while using fewer average output tokens. This advantage stems from MiniOpt's internalized unified reasoning to model and solve paradigm tailored for optimization modeling problems, which guides the model along a more efficient reasoning path during inference.

Table 9: Comparison of average output token consumption in OptMATH-Bench.

| Model | Avg. Token Count | SA (%) | ER (%) |
|---|---|---|---|
| MiniOpt-3B | 1911.72 | 42.77 | 68.67 |
| MiniOpt-7B | 3068.99 | 51.20 | 76.51 |
| DeepSeek-R1 | 5078.90 | 39.76 | 53.01 |

## L    SYSTEM PROMPT FOR CODE CONVERSION FROM GUROBIPY TO PYOMO

This section provides the system prompt for the large language model to convert GurobiPy code in OptMATH-Train into Pyomo code, where the content within [·] and {·} will be replaced with the corresponding parts.

---

**PROMPT TEMPLATE FOR CODE CONVERSION**

```
You are an expert in optimization problems. Your task is to convert
    the given gurobipy code into pyomo code.

**Instructions:**
1. Don't give any explanation, just provide the converted pyomo
    code in the following format:
```python
[pyomo code here]
```
2. Please note that the following solvers are available for use: '
    glpk', 'cbc', 'ipopt', 'scip'. Other solvers should not be
    utilized.
3. Please add 'from pyomo.environ import *' at the beginning of
    your code.
4. Please print the optimal objective value at the end of the code.

**Gurobipy code:**
{gurobipy}
```

## M  PROMPT FOR THE SOLVER SELECTION

This section provides the system prompt to guide MiniOpt models in autonomously selecting solvers after modeling optimization problems.

---

**PROMPT FOR SOLVER SELECTION**

```
**Solver Selection Guide:**
- ``glpk``: Best for small-to-medium linear problems (LP).
- ``cbc``: Recommended for mixed-integer linear programming (MILP)
    and larger linear problems. Handles binary/integer variables
    well.
- ``ipopt``: Use for nonlinear problems (NLP) with continuous
    variables. Does NOT support discrete variables.
- ``scip``: Most versatile – handles mixed-integer nonlinear
    problems (MINLP), large-scale problems, and complex constraints.

**Select solver based on:**
1. Variable types (continuous vs integer/binary)
2. Linearity of objective/constraints
3. Problem scale (small: glpk/cbc, large: scip/ipopt)
4. Nonlinearity presence (use ipopt/scip)
```

---

## N  SYSTEM PROMPT FOR RL TRAINING

This section provides the system prompt used by MiniOpt models during reinforcement learning (RL) training.

---

**SYSTEM PROMPT FOR RL TRAINING**

```
You are a helpful assistant. The assistant first thinks about the
    reasoning process in the mind and then provides the user with
    the answer. The reasoning process and answer are enclosed within
     <think> </think> and <answer> </answer> tags, respectively, i.e
    ., <think> reasoning process here </think><answer> answer here
    </answer>, please make sure to answer according to the above
    format. Now the user asks you to solve an optimization reasoning
     problem, you should:
1. Detailed reasoning about the problem within <think> </think>
    tags.
2. Write the corresponding five-element model (derived from your
    analysis).
3. Determine the mathematical properties of problem and select an
    appropriate solver from 'glpk', 'cbc', 'ipopt', 'scip'.
4. Recheck and correct if necessary at the end of the <think> </
    think> section.
   - Verify the five-element model fully captures the problem's
   requirements.
   - Confirm no constraints/variables are missing or over-
   simplified.
   - Ensure the solver choice aligns with the problem's
   mathematical properties.
5. Provide the corresponding Pyomo code based on checked five-
    element model within <answer> </answer> tags.

In mathematics, optimization problem can be modeled as the
    following expression $\\min_{{\\boldsymbol{{x}} \\in \\mathcal{{
    X}}}} f(\\boldsymbol{{x}}), {{\\rm s.t.}} G(\\boldsymbol{{x}})
    \\leq \\boldsymbol{{c}}$, where $\\boldsymbol{{x}} = (x_1, x_2,
```

---

```
        \\ldots, x_D)^\\top$ is the $D$-dimensional decision variable, $
        \\mathcal{{X}} \\subset \\mathbb{{R}}^D$ is the feasible domain,
         $f: \\mathcal{{X}} \\rightarrow \\mathbb{{R}}$ is the objective
         function and the goal is to find the minima of $f$, $G(\\
        boldsymbol{{x}}) \\leq \\boldsymbol{{c}}$ are the constraints of
         $\\boldsymbol{{x}}$.

The above definition can be mapped to a five-element consisting of
    ``Variables, Objective, Constraints, Sets, Parameters''.
    Variables indicate what $\\boldsymbol{{x}}$ is, Objective
    describes the form of the objective function $f(\\boldsymbol{{x
    }})$, and Constraints indicates the constraints $G(\\boldsymbol
    {{x}})$ and $\\mathcal{{X}}$. These three can abstract the
    optimization problem. Sets and Parameters are their specific
    explanations: Sets describe and explain the subscripts of the
    vectors or matrices in them, and Parameters supplement their
    specific values.

You need to give a detailed reasoning process for the problem first
    , and then write the corresponding five-element model based on
    the problem description and information provided by user.

Please complete the following template to model the optimization
    problem into five-element:

<think>
Your reasoning process here...

## Sets:
[You need to fill in]

## Parameters:
[You need to fill in]

## Variables:
[You need to fill in]

## Objective:
[You need to fill in]

## Constraints:
[You need to fill in]
</think>

In Pyomo, all constraints must be formulated using '<=', '>=', or
    '=='. If you need to use '>' or '<', you can introduce a very
    small value to transform the inequality. Please note that the
    following solvers are available for use: 'glpk', 'cbc', 'ipopt',
    'scip'. Other solvers should not be utilized.

**Solver Selection Guide:**
- ``glpk``: Best for small-to-medium linear problems (LP).
- ``cbc``: Recommended for mixed-integer linear programming (MILP)
    and larger linear problems. Handles binary/integer variables
    well.
- ``ipopt``: Use for nonlinear problems (NLP) with continuous
    variables. Does NOT support discrete variables.
- ``scip``: Most versatile - handles mixed-integer nonlinear
    problems (MINLP), large-scale problems, and complex constraints.

**Select solver based on:**
1. Variable types (continuous vs integer/binary)
2. Linearity of objective/constraints
```

```
3. Problem scale (small: glpk/cbc, large: scip/ipopt)
4. Nonlinearity presence (use ipopt/scip)

Please select an appropriate solver based on the type and quantity
    of variables, objectives, and constraints. After thinking, when
    you finally reach the five-element model, you should give the
    corresponding Pyomo code within the <answer> </answer> tags, i.e
    ., <answer> '''python\n code here''' </answer>. The user will
    extract the complete code you provide through the regular
    expression r"'''python\n(.*?)'''" in the <answer> </answer> tags
    . The execution result of the code should include the optimal
    solution and the objective value. The optimal objective value
    will be extracted automatically from your last printed result.
```

## O   LABELING PROMPT AND THE DATA DISTRIBUTIONS OF OPTMATH-TRAIN

This section provides the system prompts for the large language model to label the type and scenario of problems in OptMATH-Train, where $\{\{\cdot\}\}$ will be replaced with the corresponding content. After labeling, the distribution of scenarios in OptMATH-Train is displayed in Figure 6, and the distribution of types in OptMATH-Train is displayed in Figure 7

### SYSTEM PROMPT FOR TYPE LABELING

```
Please classify the following optimization problem into one of
    these technical types based on the mathematical formulation and
    decision variables, not just surface-level descriptions:

1. Linear Programming (LP): Problems with linear objective function
    and linear constraints, all continuous variables
2. Integer Programming (IP): Problems with linear or nonlinear
    components where ALL variables are discrete/integer
3. Mixed Integer Linear Programming (MILP): Problems with linear
    components containing BOTH continuous and discrete variables
4. Nonlinear Programming (NLP): Problems with nonlinear objective
    function and/or nonlinear constraints (variables may be
    continuous/discrete)
5. Combinatorial Optimization (CO): Problems focused on selecting/
    discrete structures (graphs, permutations, sets) with typically
    binary variables
6. Multi-objective Programming (MOP): Problems explicitly
    optimizing multiple conflicting objectives simultaneously
7. Second-Order Cone Programming (SOCP): Problems with a linear
    objective function, linear constraints, and second-order cone
    constraints (e.g., \(\|Ax + b\| \leq c^T x + d\))

# Problem:
{{Question}}

# Output
Analyze the mathematical structure step by step and classify its
    type. Finally, output the type abbreviation in the following
    format:
Type: Abbreviation of the type

Note:
- Focus on the fundamental mathematical formulation, not
    application domain
- Check variable types (continuous/discrete/binary) and objective/
    constraint linearity
```

```
- For MOP, there must be explicit multiple objectives
- For pure discrete problems with special structures (e.g. graphs),
    prefer CO over IP
```

---

SYSTEM PROMPT FOR SCENARIO LABELING

```
Please classify the following optimization problem into one of
    these application domains based on the core decision-making
    context and primary business function, not just keywords
    mentioned in the problem:

1. Supply Chain: Decisions about inventory management, distribution
    network, warehousing operations
2. Finance: Decisions about portfolio management, investments, risk
    management, financial planning
3. Manufacturing: Decisions about production processes, quality
    control, factory operations
4. Transportation: Decisions about routing, vehicle scheduling,
    fleet management, traffic flow, carrier selection
5. Healthcare: Decisions about medical staff scheduling, patient
    flow, hospital resources
6. Energy: Decisions about power generation, energy conservation,
    grid distribution
7. Technology: Decisions about network design, data center
    operations, cloud resources
8. Retail: Decisions about store operations, pricing, inventory,
    equipment, store layout
9. Agriculture: Decisions about farming operations, crop planning,
    irrigation
10. Logistics: Decisions about delivery operations, warehouse
    management, distribution
11. Resources: Decisions about raw materials, equipment allocation,
     material management
12. Marketing: Decisions about campaign planning, budget allocation
    , target selection
13. Education: Decisions about course scheduling, resource
    allocation in schools
14. Environment: Decisions about environmental protection,
    emissions control, conservation
15. Construction: Decisions about project planning, construction
    resource allocation
16. Military: Decisions about military operations, deployment,
    supply management
17. Sports: Decisions about game scheduling, team formation,
    strategy
18. Telecommunications: Decisions about network coverage, bandwidth
     allocation
19. Aviation: Decisions about flight scheduling, crew assignment,
    airport operations
20. Services: Decisions about service operations, staff scheduling,
     capacity management
21. Public utilities: Decisions about utility services,
    infrastructure management, service delivery
22. Other: Problems that don't clearly fit into above categories

# Problem:
{{Question}}

# Output
```

```
Let's think step by step,give the analysis of the problem and
    classify it into one of the above application domains.Finally,
    output the name of the domain in the following format:
Category: Name of the Domain

Note:
- Focus on the fundamental business function and decision-making
    context
- Don't be misled by secondary keywords or background story
- Consider who is making the decision and what is their primary
    business purpose
```

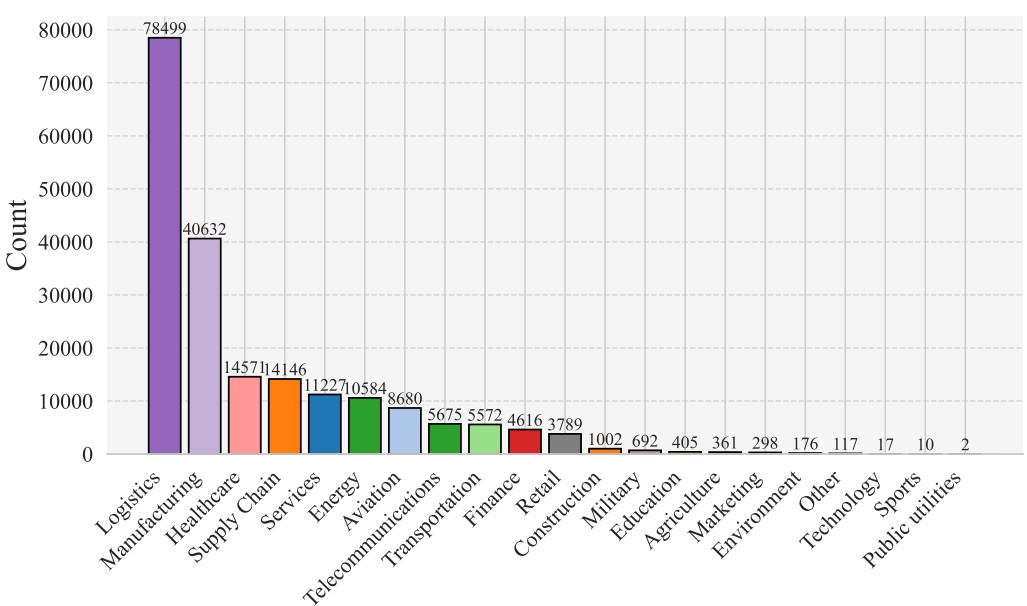

Figure 6: Proportion of every scenarios of instances in OptMATH-Train (201K).

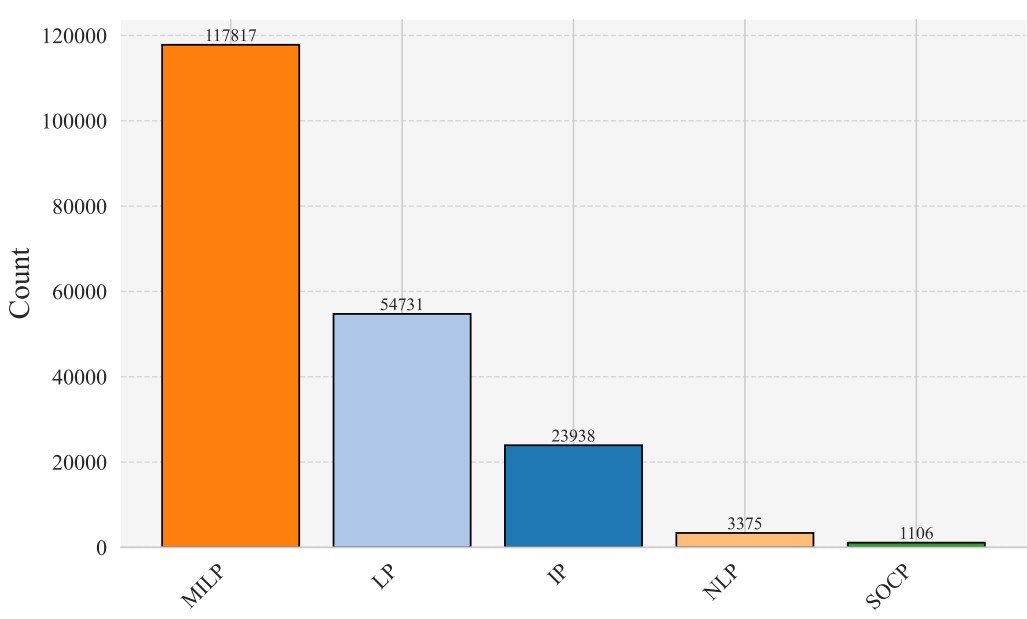

Figure 7: Proportion of every problem types of instances in OptMATH-Train (201K).

