# OpenReview forum: "MiniOpt: Reasoning to Model and Solve General Optimization Problems with Limited Resources"
_ICLR.cc/2026/Conference — ICLR 2026 Conference Withdrawn Submission_

### Official Review · Reviewer_P5FB · 2025-10-31

**Soundness:** 3
**Presentation:** 3
**Contribution:** 3
**Rating:** 6
**Confidence:** 4

**Summary:**

This paper introduces **MiniOpt**, a framework for training small-scale large language models (LLMs) to model and solve optimization problems from natural language descriptions. It addresses key challenges—data scarcity, high verification costs, and computational overhead. By proposing a two-stage reinforcement learning (RL) pipeline with a verifiable reward function (**OptReward**) and an improved RL algorithm (**OptGRPO**). MiniOpt employs a structured “reasoning to model and solve” paradigm, where the model first formulates problems using a five-element representation and then generates executable Pyomo code. Evaluated across nine benchmarks, MiniOpt-3B outperforms many larger models, demonstrating strong optimization generalization with significantly lower resource requirements, making it suitable for local deployment in resource-constrained environments.

**Strengths:**

1.  **Strong Performance with High Efficiency:** MiniOpt achieves state-of-the-art or highly competitive solving accuracy (SA) on multiple benchmarks, even with its small 3B parameter size, outperforming many larger models like GPT-5 and LLMOPT-14B. Crucially, it does so while consuming significantly fewer computational tokens, making it highly efficient and cost-effective for local deployment.

2.  **Comprehensive and Rigorous Experiments:** The paper validates its claims through extensive experiments across nine diverse optimization benchmarks, spanning various problem types and real-world scenarios. This thorough assessment, complemented by detailed ablation studies, robustly demonstrates the model's generalization capability and the contribution of each proposed component.

3.  **A Novel, Resource-Conscious Training Framework:** The core contribution is a meticulously designed pipeline that effectively trains small-scale LLMs under limited resources. The integration of a two-stage RL process, a verifiable reward function (OptReward), and strategic data selection successfully overcomes the data-hungry and computationally intensive nature of traditional methods for this complex task.

**Weaknesses:**

### **Weaknesses 1**
A significant limitation is the framework's tight coupling with the **GurobiPy/Pyomo ecosystem** during both training and evaluation. The SFT data is derived from GurobiPy conversions, and the output is constrained to Pyomo code targeting a limited set of solvers. This design lacks **solver-agnostic flexibility**. A model might correctly understand a problem's structure but fail due to Pyomo-specific syntax or the selected solver's limitations, while an alternative solver or modeling language (e.g., CVXPY, PuLP, or PySCIOPT) could have succeeded. This restricts the generalizability and practical utility of the approach across the diverse landscape of optimization tools.


### **Weaknesses 2**
Another limitation is the **lack of diagnostic metrics** to disentangle the root causes of failure. The final reward and accuracy scores conflate two distinct types of errors:
1.  **Conceptual Modeling Errors:** The model fails to reason correctly and produces an **inaccurate mathematical formulation**.
2.  **Solver Programming Errors:** The model understands the problem but generates syntactically incorrect or semantically flawed Pyomo/Gurobi code.

Without a separate, fine-grained evaluation of the five-element formulation's correctness *before* code generation, it is impossible to diagnose whether failures stem from a lack of reasoning or merely implementation incompetence. This obscures the true bottleneck for model improvement.

**Questions:**

1.  **Error Diagnosis:** Can you break down where the model fails most? Is it primarily in the *mathematical reasoning* (producing a wrong formulation) or in the *code implementation* (writing bad Pyomo code)?

2.  **Generality:** Is your model's performance tied to Pyomo? Would it fail if asked to generate code for a different optimization tool like CVXPY, or has it learned a general skill for modeling optimization problems?

---

> ### Author Response · Authors · 2025-11-24
> **Reply to Reviewer P5FB (1/2)**
>
> ### Response to Question 1: About the Verification of Process
> Thank you for your concerns about the verification of modeling process! We have carefully considered your feedback and provided detailed responses below.
>
> To determine the cause of an error in a sample that failed to solve involves process validation. **More precise process verification methods[1] or verification approaches targeting modeling results [2,3] tend to incur higher costs, since they often involve process reward models or complex verification algorithms. To maintain low training and inference costs, this paper adopts the simpler RLVR approach**, designing reward functions with minimal verification overhead to validate structural completeness of modeling and the final results. This method is widely used in the field of reasoning models [4]. Regarding the correctness of the modeling process, this study can accurately verify whether the modeling structure is complete, though semantic correctness cannot be directly validated. To determine the causes of failure in unsuccessful solving samples, two methods can be considered:
>
> 1) Manual inspection of the content within the `<think>` tags, or automated evaluation using models such as GPT-4o or Deepseek-V3 via the LLM-as-a-Judge approach.
> 2) Rough estimation: Let SA denote the model's solving accuracy and ER denote the proportion of samples where code execution succeeds. **Based on inference outcomes, 1 - ER represents the proportion of samples where code execution fails, while ER - SA indicates the proportion of samples where code runs successfully but produces incorrect results**. These two metrics can roughly reflect errors in code implementation and modeling steps, respectively. (ER - SA) + (1 - ER) = 1 - SA indicates that these two error calculation methods comprehensively cover all cases of solving failures without double-counting erroneous samples.
>
> For example, on OptiBench, MiniOPT-3B achieves SA = 53.55% and ER = 83.64%. Among all failed samples (1 - SA = 46.45%), 16.36% (1 - ER) are directly caused by code implementation errors, while 30.09% (ER - SA) are due to modeling errors.
>
>
> [1] BPP-Search: Enhancing Tree of Thought Reasoning for Mathematical Modeling Problem Solving, https://aclanthology.org/2025.acl-long.40/
>
> [2] EquivaMap: Leveraging LLMs for Automatic Equivalence Checking of Optimization Formulations, https://openreview.net/forum?id=RvdjzNlksm
>
> [3] ORGEval: Graph-Theoretic Evaluation of LLMs in Optimization Modeling, https://arxiv.org/abs/2510.27610
>
> [4] Logic-RL: Unleashing LLM Reasoning with Rule-Based Reinforcement Learning, https://proceedings.neurips.cc/paper_files/paper/2023/file/271db9922b8d1f4dd7aaef84ed5ac703-Paper-Conference.pdf

---

> > ### Author Response · Authors · 2025-11-24
> > **Reply to Reviewer P5FB (2/2)**
> >
> > ### Response to Question 2
> > Thank you for your careful review. We have carefully considered your feedback and provided detailed responses below. If there are any other questions, please feel free to ask, and we will respond promptly.
> >
> > (1) Use of Pyomo in this work: Since widely-used packages like Gurobi and COPT are close-source, generating code for them may entail licensing costs, and a model's generalization capability across different modeling languages and solvers depends on the proportion of relevant corpora in its pre-training and post-training data. Therefore, following established practices [1], we adopted the open-source, solver-agnostic Pyomo modeling language as the language for our training and inference solver code.
> >
> > (2) Regarding general solving capability: One of the key consideration is whether the selected solver can handle as many problem types as possible. For example, the GLPK solver does not support nonlinear optimization problems. If applied to such problems, it will lead to incorrect results. **For MiniOPT, we aimed to utilize the open-source Pyomo package while ensuring compatibility with multiple open-source solvers to maximize problem coverage and meet diverse solving requirements.** MiniOPT dynamically selects the most suitable solver from four open-source options (this study employed 'glpk', 'cbc', 'ipopt', and 'scip'). The applicability of these four solvers are as follows:
> >   - glpk: Optimal for small- to medium-scale linear programming problems.
> >   - cbc: Recommended for mixed-integer linear programming (MILP) and larger-scale linear problems. Handles binary/integer variables effectively.
> >   - ipopt: Suitable for nonlinear programming (NLP) problems with continuous variables. Does not support discrete variables.
> >   - scip: The most versatile solver, capable of handling mixed-integer nonlinear programming (MINLP), large-scale problems, and complex constraints.
> >
> > Therefore, we are confident that MiniOPT's capability to autonomously select appropriate solvers based on problem characteristics represents a significant advantage. **Experimental results in Section 4.2 (Breadth across Types and Scenarios) indicate that MiniOPT achieves generalized performance across multiple optimization problem types and diverse application scenarios**.
> >
> > [1] LLMOPT: Learning to Define and Solve General Optimization Problems from Scratch, https://openreview.net/forum?id=9OMvtboTJg

---

> ### Author Response · Authors · 2025-11-28
> **Gentle Reminder of the Rebuttal Deadline**
>
> Dear Reviewer P5FB,
>
> We have carefully revisited your question and provide explanations from the following two perspectives in the hope of addressing your concerns.
>
> **Process Verification**: This study focuses on training models with reinforcement learning at lower verification costs. For failed problem-solving cases, the content within the `<think>` tags can be inspected either manually or via LLMs. Alternatively, predictions can be made based on whether code execution succeeds and whether the results are correct.
>
> **Modeling Language and Solver**: MiniOPT relies on the open-source modeling language Pyomo for code generation and **is trained to adaptively select the most suitable open-source solver** based on specific types of optimization problems，including 'glpk', 'cbc', 'ipopt', and 'scip'. Experimental results in Section 4.2 demonstrate that MiniOPT exhibits strong generalization capabilities across diverse optimization problem types.
>
> As the deadline approaches, we sincerely hope to address your concerns further. We appreciate your effort and constructive comment once again!
>
> Best regards,
>
> Authors

---

### Official Review · Reviewer_t9Ho · 2025-10-31

**Soundness:** 2
**Presentation:** 3
**Contribution:** 2
**Rating:** 4
**Confidence:** 4

**Summary:**

This paper proposes a training and evaluation framework for learning to model natural language and solve optimization problems with small-scale LLMs under resource constraints. The main methods include (1) structuring the task into a paradigm with <think>...</think> (five-element model) and <answer>...</answer> (executable Pyomo code); (2) designing a verifiable reward function OptReward (format/five elements/numerical accuracy) to ensure stable output of the five elements through a two-stage RL training process; (3) selecting a suitable RL algorithm from the existing GRPO trick.

**Strengths:**

1.  A clear paradigm (<think> five elements + <answer> executable scripts) with verifiability, reducing the labor cost of annotation and verification.
2.  Detailed experimental analysis (demonstrating the contributions of modules such as SFT warm-up, OptReward, and OptGRPO).
3.  Advantages in practical evaluation metrics (SA, ER, and token cost).

**Weaknesses:**

1. The approach lacks innovation, as it merely enables the LLM to produce a five-element structure consistent with the verification format. Such verifiable schemes are fairly common in LLM optimization.
2. It does not offer significant improvements to the GRPO algorithm; rather, it incorporates a few well-established techniques from previous GRPO studies.
3. The paper failed to compare its approach with several closely related studies. Notable omissions include: [1] ORLM: A Customizable Framework in Training Large Models for Automated Optimization Modeling. (2025). [2] Step-Opt: Boosting Optimization Modeling in LLMs through Iterative Data Synthesis and Structured Validation (2025).

**Questions:**

1. In the method description (Figure 1), the upper bound for gradient clipping in the RL algorithm is incorrectly specified. The formula should use a '+' symbol instead of a '–'."
2.  Could the authors provide ablation studies to demonstrate the impact of removing KL divergence and asymmetric clipping on training stability and performance?
3.  The paper would benefit from a performance comparison against other competitive learning-based LLM models for optimization modeling and solving.

---

> ### Author Response · Authors · 2025-11-24
> **Reply to Reviewer t9Ho (1/3)**
>
> ### Response to Weakness 1: The role of five-element modeling
> Thank you for your careful review! We are glad that you paid attention to the modeling approach in this paper. In fact, the contribution of this work extends beyond proposing a novel verification paradigm, it introduces an efficient training pipeline for operations optimization problems, which **achieves strong optimization generalization capabilities with limited data and small-scale models**. We will now address your concerns regarding the innovation of this paper:
>
> > 1) **To enable stable training of operations optimization models with limited data**, we designed a Two-Stage RL approach. The model first learns the "Reasoning to model and solve" paradigm on simpler data, and then proceeds to train on a carefully selected subset of high-difficulty data with diverse problem types and scenarios. **This process enhances the model's generalization capability in optimization tasks**.
> > 2) **To align with the reasoning pattern of five-element modeling and improve the model's ability to solve challenging problems**, we modified the GRPO algorithm by removing the KL divergence term and adjusting the clipping upper bound. This encourages the model to more extensively explore solution strategies within the structured five-element format. To enhance sampling efficiency with limited data, we adopt a token-level loss to obtain dense rewards in long outputs.
> > 3) Finally, regarding verification methods, previous studies have employed modeling approaches similar to the five-element framework. For instance, OptiMUS [1] utilizes LLM-as-a-Judge to inspect states throughout the modeling process, while BPP-Search [2] employs a process reward model to evaluate each element of the modeling procedure. Although these methods are effective, they incur high computational costs during either inference or training. **The key innovation of our work lies in leveraging the five-element structure as a foundation for designing reward functions, thereby enabling low-cost verification that effectively supports RLVR paradigm**.
>
> ### Response to Question 1: Corrected a typographical error
> We have corrected the formula in the method section of the figure (changing the symbol before $\varepsilon_{high}$ from '-' to '+') and replaced the correct figure in the paper.
>
> [1] Ahmaditeshnizi, A., Gao, W. &amp; Udell, M.. (2024). OptiMUS: Scalable Optimization Modeling with (MI)LP Solvers and Large Language Models. <i>Proceedings of the 41st International Conference on Machine Learning</i>, in <i>Proceedings of Machine Learning Research</i> 235:577-596 Available from https://proceedings.mlr.press/v235/ahmaditeshnizi24a.html.
>
> [2] Teng Wang, Wing Yin Yu, Zhenqi He, Zehua Liu, HaileiGong HaileiGong, Han Wu, Xiongwei Han, Wei Shi, Ruifeng She, Fangzhou Zhu, and Tao Zhong. 2025. BPP-Search: Enhancing Tree of Thought Reasoning for Mathematical Modeling Problem Solving. In Proceedings of the 63rd Annual Meeting of the Association for Computational Linguistics (Volume 1: Long Papers), pages 821–838, Vienna, Austria. Association for Computational Linguistics.

---

> > ### Author Response · Authors · 2025-11-24
> > **Reply to Reviewer t9Ho (2/3)**
> >
> > ### Response to Question 2: Motivation for improving GRPO
> > Thank you for your valuable suggestions. We would like to clarify that OptGRPO is a algorithm designed for the MiniOPT training pipeline, which incorporates a two-stage RL framework and OptReward. We apologize for any confusion caused by our original description and have revised the relevant expressions in the Section 3.4.2 and highlighted in blue to prevent misunderstandings.
> >
> > Regarding the impact of removing KL divergence and asymmetric clipping on performance, **we provide ablation studies in Table 2. As shown, using standard GRPO leads to a significant performance drop on MiniOPT-3B, with an average accuracy decrease of 8.61%**. This indicates that the removal of KL divergence and asymmetric clipping effectively promotes the model’s exploration of new policies, encouraging it to extensively explore high-reward responses under the five-element paradigm designed for solving OR problems. Additionally, due to our two-stage RL training process—where the first stage teaches the model the response paradigm based on the five-element structure, and the second stage encourages extensive exploration on challenging problems—this pipeline also contributes to improved training stability. **As demonstrated in Table 2, removing the two-stage RL results in a performance decline of 2.82%.**

---

> ### Author Response · Authors · 2025-11-24
> **Reply to Reviewer t9Ho (3/3)**
>
> ### Response to Question 3: New experiments about Step-OPT and ORLM
> Thank you for your valuable suggestion regarding additional comparative experiments. We have incorporated reproduced experiments of Step-OPT[1] and evaluated its performance on the same 9 benchmarks used in this paper. The training of Step-OPT follows the configuration in the original paper: 12 epochs of LoRA-based SFT with LLaMA as base model are performed using the same training dataset as in [1]. To align with the MiniOPT in this study, we also trained models with 7B and 3B parameters using Qwen2.5 as the base model and evaluated their SA. We calculated three sets of experimental results, as shown in the table below. The inference prompt also adheres to the settings in [1], and the benchmarks used are consistent with those in this study (MiniOPT). It should be noted that the benchmarks evaluated in the original Step-OPT paper include NL4OPT, Mamo_Easy, Mamo_Complex, and IndustryOR. These benchmarks were processed and refined by Step-OPT-Instruct to improve data quality, and therefore differ from the benchmarks evaluated in this study, resulting in reasonable variations in performance.
>
>    Baselines| Avg. |NL4OPT | ICML.C | Mamo_Easy | Mamo_Complex | NLP4LP |   ComplexOR|IndustryOR  | OptiBench | OptMATH
> ------------|-------|-------|--------|-----------|--------------|--------|--------|--------|--------|--------
> Step-OPT-LLaMA-3.2-3B | 0.3961|0.7000  |0.2659  |0.6810     | **0.3649**       |0.6364  |  0.3889    |0.1700      |0.2678     | 0.0904
> Step-OPT-Qwen2.5-3B | 0.3615 | 0.4130 | 0.3854 | 0.7531 | 0.2085 | 0.5331 | 0.2778 | **0.2100** | 0.4000 | 0.0723
> MiniOPT-Qwen2.5-3B|**0.5694**|**0.8304**    |**0.6805**  |**0.8543**    |0.3507        |**0.7355**    |**0.5000**    |**0.2100**   |**0.5355**   |**0.4277**
> Step-OPT-LLaMA3-8B | 0.5080|0.7522  |0.6854  |0.7945     | **0.5071**       |0.6488  |  0.2778    |**0.2700**      |0.4975     | 0.1386
> Step-OPT-Qwen2.5-7B | 0.4742 | 0.7783 |0.5732 | 0.6933 | 0.5024 | 0.4835 | 0.3889 | **0.2700** | 0.4876 | 0.0904
> MiniOPT-Qwen2.5-7B |**0.6276**|**0.8913**   |**0.7756**   |**0.8834**   |0.3839     |**0.7934**   |**0.5556**   |0.2600     |**0.5934**   |**0.5120**
>
> The results from the table show that MiniOPT has significant advantages.
>
> As for ORLM[2], **since we are unable to access its original training dataset, it is difficult to reproduce ORLM, thus we cite the experimental results reported in the ORLM paper[2] across 4 benchmarks**. The comparative results between ORLM and MiniOPT-3B and MiniOPT-7B are summarized in the table below:
>
> Baselines |NL4OPT	|MAMO_Easy	|MAMO_Complex|	IndustryOR
> -----------------|----------|-----------|------------|-------
> ORLM-Qwen2.5-7B  | 86.10%	|  85.20%   |	44.10%	 |	25.00%
> MiniOPT-7B       |89.13%    |	88.34%  |   38.39%   |26.00%
> MiniOPT-3B       |83.04%    |  85.43%   |    35.07%  |21.00%
>
> Experimental results indicate that MiniOPT-3B performs competitively with ORLM, while MiniOPT-7B shows a significant advantage over ORLM.
>
> [1] Training LLMs for Optimization Modeling via Iterative Data Synthesis and Structured Validation. https://aclanthology.org/2025.findings-emnlp.691/
>
> [2] Chenyu Huang; , Zhengyang Tang; , Shixi Hu; , Ruoqing Jiang; , Xin Zheng; , Dongdong Ge; , Benyou Wang; , Zizhuo Wang (2025) ORLM: A Customizable Framework in Training Large Models for Automated Optimization Modeling. Operations Research 0(0). https://doi.org/10.1287/opre.2024.1233

---

> ### Author Response · Authors · 2025-11-28
> **Gentle Reminder of the Rebuttal Deadline**
>
> Dear Reviewer t9Ho,
>
> We have carefully revisited your question and provide explanations from the following two perspectives in the hope of addressing your concerns.
> The five-element framework serves to provide a chain-of-thought structure for the Two-Stage RL, thereby **constraining reasoning to higher-score regions** and playing a important role in enhancing MiniOPT's generalization capability. The three improvements in OptGRPO are specifically **designed to align with MiniOPT's training pipeline**, enhancing the model's ability to explore low-probability tokens under the constraints of the Five-Element framework while improving sampling efficiency with limited data. Experimental results in Table 2 of the ablation studies (Section 4.4) validate the contributions of these two aspects. Furthermore, following your suggestion, we have added comparative experiments involving Step-OPT and ORLM.
>
> As the deadline approaches, we sincerely hope to address your concerns further, if you find our response satisfactory, could you please kindly consider the possibility of updating the rating. We appreciate your effort and valuable feedback once again!
>
> Best regards,
>
> Authors

---

### Official Review · Reviewer_Etef · 2025-10-31

**Soundness:** 3
**Presentation:** 3
**Contribution:** 2
**Rating:** 4
**Confidence:** 5

**Summary:**

This paper proposes MiniOpt, a framework to train small-scale LLMs (e.g., 3B) to solve optimization problems under limited resources. The method uses a three-stage training pipeline (SFT Warm-up, RL Stage-1, and RL Stage-2). This pipeline is guided by a verifiable reward function (OptReward) and a RL algorithm (OptGRPO). Experiments show the MiniOpt-3B model achieves solving accuracy competitive with models larger than 10B parameters

**Strengths:**

1. **Strong Performance-to-Parameter Ratio:** The paper demonstrates a strong empirical performance-to-parameter trade-off. The 3B model, MiniOpt-3B, achieves an average solving accuracy of 56.94%. This result is shown to be competitive with several models in the >10B class.

2. **Reported Inference Efficiency:** The method shows a clear advantage in inference cost, aligning with its primary goal of low-cost deployment. On the challenging OptMATH-Bench, MiniOpt-3B is reported to achieve a higher SA than DeepSeek-R1 while consuming fewer average output tokens.

3. **Systematic Training Pipeline and Reward Design:** The paper contributes a complete and structured training pipeline, beginning with an SFT warm-up to mitigate sparse rewards, followed by a two-stage RL process for paradigm acquisition and generalization .

**Weaknesses:**

1. **Stated Motivation Lacks Novelty:** The paper's primary motivation—addressing data privacy and deployability through small, open-source models—is not a new contribution to this specific research area. This motivation was already a core premise for foundational work like ORLM and has been a shared consideration in subsequent studies such as OptMATH , ReSocratic, and SIRL . Most of these works train 7B-8B models, and MiniOpt's 3B variant represents only a quantitative reduction in scale rather than a fundamentally different problem setting. Therefore, the problem framing does not offer a novel perspective or tackle a new challenge relative to the established literature.

2. **Contradictory Claims Regarding "Limited Resources":** The paper's premise is critically undermined by a direct contradiction. It claims to operate under "limited resources" and highlights the difficulty of acquiring high-quality data. However, the crucial SFT Warm-up stage relies on the OptMATH-Train dataset , which the paper states contains more than 100K samples. This is a massive dataset, far exceeding the ~30K samples used in related works like ORLM and ReSocratic. This methodology is not "resource-limited" and directly conflicts with the paper's central motivation.

3. **Incomplete Treatment of Related Work:** The paper omits critical comparisons with relevant work:
(1) Step-Opt[1]: This work also addresses stepwise optimization modeling with 7B-8B models but is not cited.
(2) Missing DAPO comparison: The paper states OptGRPO "builds upon" prior improvements like DAPO but provides no experimental comparison with DAPO itself.

[1] Wu Y, Zhang Y, Wu Y, et al. Step-Opt: Boosting Optimization Modeling in LLMs through Iterative Data Synthesis and Structured Validation[J]. arXiv preprint arXiv:2506.17637, 2025.

**Questions:**

1. Regarding the paper's motivation, the focus on deployability and privacy for small models is shared by prior work (ORLM, SIRL, etc.). Could you clarify what you see as the novel problem setting or challenge that MiniOpt addresses, beyond a quantitative reduction in model scale from the established 7B/8B range to 3B?

2. This paper is premised on operating under "limited resources," yet the SFT Warm-up stage relies on the 140K-sample OptMATH-Train dataset. This appears to be a much larger data requirement than related works. Could you clarify this apparent contradiction and explain how this massive SFT dataset aligns with the "limited resources" claim?

3. Please include a baseline comparison against DAPO to experimentally validate the algorithmic contribution.

---

> ### Author Response · Authors · 2025-11-24
> **Reply to Reviewer Etef (1/3)**
>
> ### Response to Question 1: The key contribution of this paper.
>
> We appreciate your valuable and thoughtful feedback. We would like to clarify that our primary contribution lies in proposing a training pipeline with lower costs, which achieves competitive or even superior performance while training models with fewer parameters, using fewer resources and smaller datasets. The reduction in model scale is a direct outcome of this pipeline.
>
> Its advantages are as follows:
> > 1) By integrating this Two-stage RL training pipeline with a specifically designed OptReward function, the OptGRPO algorithm, and data selection strategies, the RLVR-based reasoning paradigm enables stronger modeling and solving capabilities, as well as robust optimization generalization. Due to the low cost of this training pipeline, spanning from data acquisition to the entire training process, it effectively addresses the challenge of private deployment.
>
> > 2) To highlight the comparative advantages with similar models, we have added experiments by reproducing Step-OPT. We first followed the original paper[1] to train models using LLaMA as the base model with 8B(LLaMA-3) and 3B(LLaMA-3.2), then evaluated their Solution Accuracy (SA). To align with the MiniOPT in this study, we also trained models with 7B and 3B parameters using Qwen2.5 as the base model and evaluated their SA. We calculated three sets of experimental results and determined the performance degradation as the parameter scale decreased, as shown in the table below. Step-OPT-LLaMA-3-8B achieves an average SA of 50.80% across 9 benchmarks, while the proposed MiniOPT attains an average SA of 62.76%. **When the parameter scale of both models are reduced to Step-OPT-LLaMA-3.2-3B, Step-OPT-LLaMA exhibit a performance drop of 11.19% from 8B to 3B. For Step-OPT-Qwen2.5, the SA degradation is 11.27%, while MiniOPT only deceased by 5.82%**. This indicates that the key advantage of MiniOPT lies in its ability to maintain superior performance even with reduced parameter scale. From the perspective of capability density [2], MiniOPT effectively achieves lower parameter requirements and inference costs while preserving comparable performance.
>
> Model                |  Solution Accuracy  | $\Delta $SA
> ---------------------|---------------------|------------
> Step-OPT-LLaLA-3-8B  |50.80%| 0
> Step-OPT-LLaLA-3.2-3B|39.61%|-11.19%
> Step-OPT-Qwen2.5-7B  |47.42%|0
> Step-OPT-Qwen2.5-3B  |36.15%|-11.27%
> MiniOPT-Qwen2.5-7B   |**62.76%**|0
> MiniOPT-Qwen2.5-3B   |**56.94%**|**-5.82%**
>
> $\Delta$ SA represents the difference in SA after the model parameter size decreases.
>
> > 3) Approaches like ORLM, OptMATH, and Step-OPT depend on large-scale synthetic data to improve performance through SFT, which leads to higher data and training costs. Its generalization is also limited by the distribution of the generated data. Compared with them, our method is more efficient and achieves strong optimization generalization with only a small amount of data.
>
> [1] Training LLMs for Optimization Modeling via Iterative Data Synthesis and Structured Validation. https://aclanthology.org/2025.findings-emnlp.691/
>
> [2] Densing Law of LLMs. https://www.nature.com/articles/s42256-025-01137-0

---

> ### Author Response · Authors · 2025-11-24
> **Reply to Reviewer Etef (2/3)**
>
> ### Response to Question 2: The role of SFT warm-up.
> Thank you for your concerns about the training pipeline in this paper! We would like to clarify the role of SFT warm-up with you, and if there are any other questions, please feel free to ask, and we will respond promptly.
>
> > 1) The objective of incorporating SFT Warm-up is solely to **equip the model with basic problem-solving abilities in operations optimization**, if a base model inherently possesses strong capabilities of optimization modeling and solving, the Warm-up step may be deemed unnecessary. Thus, we limited SFT Warm-up to only 1 epoch, significantly reducing resource consumption compared to approaches like OptMATH, which trains for 3-5 epochs on 200K data.
> > 2) Second, in terms of its impact on final model performance: We emphasize that the performance improvement of MiniOpt does not primarily rely on the 140K cold-start samples. **As demonstrated in the ablation studies of Table 2 in Section 4.4, SFT Warm-up with 140K data contributes only a +3.35% performance gain, whereas the two-stage RL training, utilizing a total of 4.5K data, contributes a +17.69% improvement.** This indicates that the two-stage RL approach proposed in this study for operations optimization is the key factor enhancing model capability.
> > 3) Futhermore, from the perspective of data acquisition cost: The 140K cold-start data are entirely sourced from open-access public datasets. In contrast, the training of [1,2] relies heavily on high-quality CoT data, incurring substantial distillation and annotation costs. Since our SFT Warm-up only aims to establish foundational problem-solving capabilities in operations optimization, it does not require costly high-quality CoT data. Thus, the acquisition cost of this 140K cold-start data is minimal, aligning with our study’s focus on maximizing operations optimization performance of small-scale parameter models under limited resources.
>
> [1] LLaMoCo: Instruction Tuning of Large Language Models for Optimization Code Generation, https://arxiv.org/abs/2403.01131
>
> [2] LLMOPT: Learning to Define and Solve General Optimization Problems from Scratch, https://openreview.net/forum?id=9OMvtboTJg

---

> ### Author Response · Authors · 2025-11-24
> **Reply to Reviewer Etef (3/3)**
>
> ### Adding new Experiments about Step-OPT
> Thank you for your suggestion! We have added new experiment to explain the advantages of MiniOPT over StepOPT, OptGRPO over DAPO, and revised it in the latest version of the paper.
>
> Experimental setting about Step-OPT: The training of Step-OPT follows the configuration in the original paper [1]: 12 epochs of LoRA-based SFT with LLaMA as base model are performed using the same training dataset as in [1]. To align with the MiniOPT in this study, we also trained models with 7B and 3B parameters using Qwen2.5 as the base model and evaluated their SA. We calculated three sets of experimental results, as shown in the table below. The inference prompt also adheres to the settings in [1], and the benchmarks used are consistent with those in this study (MiniOPT). It should be noted that the benchmarks evaluated in the original Step-OPT paper include NL4OPT, Mamo_Easy, Mamo_Complex, and IndustryOR. These benchmarks were processed and refined by Step-OPT-Instruct to improve data quality, and therefore differ from the benchmarks evaluated in this study, resulting in reasonable variations in performance.
>
>    Baselines| Avg. |NL4OPT | ICML.C | Mamo_Easy | Mamo_Complex | NLP4LP |   ComplexOR|IndustryOR  | OptiBench | OptMATH
> ------------|-------|-------|--------|-----------|--------------|--------|--------|--------|--------|--------
> Step-OPT-LLaMA-3.2-3B | 0.3961|0.7000  |0.2659  |0.6810     | **0.3649**       |0.6364  |  0.3889    |0.1700      |0.2678     | 0.0904
> Step-OPT-Qwen2.5-3B | 0.3615 | 0.4130 | 0.3854 | 0.7531 | 0.2085 | 0.5331 | 0.2778 | **0.2100** | 0.4000 | 0.0723
> MiniOPT-Qwen2.5-3B|**0.5694**|**0.8304**    |**0.6805**  |**0.8543**    |0.3507        |**0.7355**    |**0.5000**    |**0.2100**   |**0.5355**   |**0.4277**
> Step-OPT-LLaMA3-8B | 0.5080|0.7522  |0.6854  |0.7945     | **0.5071**       |0.6488  |  0.2778    |**0.2700**      |0.4975     | 0.1386
> Step-OPT-Qwen2.5-7B | 0.4742 | 0.7783 |0.5732 | 0.6933 | 0.5024 | 0.4835 | 0.3889 | **0.2700** | 0.4876 | 0.0904
> MiniOPT-Qwen2.5-7B |**0.6276**|**0.8913**   |**0.7756**   |**0.8834**   |0.3839     |**0.7934**   |**0.5556**   |0.2600     |**0.5934**   |**0.5120**
>
> The results from the table show that MiniOPT has significant advantages.
>
> [1] Training LLMs for Optimization Modeling via Iterative Data Synthesis and Structured Validation. https://aclanthology.org/2025.findings-emnlp.691/
>
> ### Clarification on OptGRPO and DAPO
>
> Thank you for your careful review! We apologize that the introduction of OptGRPO is confusing. We have added the following ablation experiments about DAPO, and we will carefully revise the description of OptGRPO in the paper.
>
> Setting                | SA  | ER
> -----------------------|------| ----------
> MiniOPT-3B with OptGRPO| **56.94%**  | **88.04%**
> MiniOPt-3B with DAPO   |  54.57% | 86.82%
> MiniOPt-3B with GRPO   |  48.33% | 80.00%
>
> SA represents the average solution accuracy of the model on all benchmarks, and ER represents the average execution rate of the model on all benchmarks. Replacing OptGRPO with standard DAPO results in SA drop of 2.37% and ER drop of 1.22%.
>
> [2] DAPO: An Open-Source LLM Reinforcement Learning System at Scale, https://arxiv.org/abs/2503.14476

---

> ### Author Response · Authors · 2025-11-28
> **Gentle Reminder of the Rebuttal Deadline**
>
> Dear Reviewer Etef,
>
> As the deadline approaches, we sincerely hope that your concerns have been addressed. If you have any questions, please feel free to ask directly!
>
> We have provided detailed responses regarding **our motivations and the role of SFT**. Additionally, **we have supplemented the discussion with comparative experiments between MiniOPT and Step-OPT at different parameter scales** to illustrate the core problems addressed in this paper. The results demonstrate that MiniOPT, trained under our proposed pipeline, achieves stronger optimization generalization capabilities with smaller parameter counts. Furthermore, we have clarified **the design rationale behind OptGRPO and its performance improvements over standard GRPO**.
>
> The specific details have been thoroughly discussed in our previous responses, if you are satisfied with our reply, we would greatly appreciate it if you could consider adjusting the rating.
>
> We sincerely appreciate your effort and constructive comment once again!
>
> Best regards,
>
> Authors

---

### Author Response · Authors · 2025-11-24
**General Response to Reviewers and Revision Submitted**

We would like to express our gratitude to the reviewers for their valuable feedback and insightful suggestions. We are greatly encouraged by their recognition of our proposed training pipeline as systematic (**Reviewer Etef**) and resource-conscious (Reviewer P5FB), the ''reasoning to model and solve paradigm'' as efficient ( **Reviewer t9Ho**), and the proposed model as high-performance (**Reviewer Etef**,  **Reviewer t9Ho**,  **Reviewed P5FB**). Additionally, we are pleased that the reviewers acknowledged the design of our experiments and the results (**Reviewer Etef**,  **Reviewer t9Ho**,  **Reviewer P5FB**), as well as their attention to our detailed discussions.

Below, we provide a summary of the key modifications:

+ We included additional clarifications in the section 4.1 (experimental setup) concerning the **selected modeling language software packages and solvers**.(**Reviewer P5FB**)
+ We have added **new comparison experiments (Step-OPT with different base models and different parameter scales)** in Section 4.2, **new ablation experiments (DAPO)** in Section 4.4, and corresponding analysis. Then update the related figures and supplement the corresponding methods descriptions to Appendix B.(**Reviewer Etef**, **Reviewer t9Ho**)
+ Section 3.3.1 are revised to clarify the motivation of SFT warm-up(**Reviewer Etef**), and Section 3.4.2 to the OptGRPO(**Reviewer t9Ho**).
+ A discussion is added in Section 4.3 regarding the ability of  performance retention of model when parameter reduction is implemented.(**Reviewer Etef**)

**We have made these revisions to the manuscript.** Several minor grammars errors and typos are corrected. All changes have been highlighted in blue in the manuscripts.

---

### Author Response · Authors · 2025-12-03
**A Summary of Rebuttal Results to Area Chair**

**Dear AC,**

Thank you for handling this review process for us. We have summarized the key points of this round of discussion and the revisions we have made, hoping it can provide you with reference value for making the final assessment.

During the discussion phase, each reviewer provided constructive feedback. **The reviewers give positive evaluations and recognition regarding** **the design of our training framework** ('Reviewer Etef', 'Reviewer t9Ho', 'Reviewer P5FB'), **experiments design** ('Reviewer t9Ho', 'Reviewer P5FB'), **performance-scale trade-off results** ('Reviewer Etef', 'Reviewer P5FB'), and **inference performance** ('Reviewer Etef', 'Reviewer t9Ho'). Simultaneously, they provided the following suggestions in this round of discussion:

> 1) Clarify the problem addressed by reducting the model’s parameter scale. ('Reviewer Etef')
> 2) Explain the role of SFT warm-up and OptGRPO in the training pipeline. ('Reviewer Etef', 'Reviewer t9Ho')
> 3) Explain why MiniOPT adopts Pyomo as the modeling language rather than others (e.g., Gurobi, CVXPY), and whether it is compatible with solvers beyond those used in this study. ('Reviewer P5FB')
> 4) Additional baseline experiments (Step-OPT, ORLM) and ablation studies (GRPO, DAPO) are needed. ('Reviewer Etef', 'Reviewer t9Ho')

**Based on these suggestions, we have clarified the questions raised, supplemented and explained the required new experiments, and revised the manuscript accordingly into a new version. Please refer to the discussion history on this page for details.** A summary of the revisions is provided below:

> 1) **New comparative experiments**: We updated results for the new baseline methods Step-OPT, conducted comparative experiments across different base models and parameter scales, while also citing the original results from ORLM for comparison.
> 2) **New ablation experiments**: An ablation study replacing OptGRPO with DAPO/GRPO have been added, along with corresponding discussion.
> 3) **Explanation of modeling language and solver selection**: We clarified the rationale for choosing Pyomo as the modeling language and explained how solver adaptation during training enables solving different types of optimization problems.
> 4) **Explanation of the training pipeline and its components**: We emphasized **the low-cost training pipeline designed for small-scale models with limited data**, clarified the roles of SFT warm-up and OptGRPO in the pipeline, and highlighted the importance of the Two-Stage RL. To achieve high performance with limited data and smaller model sizes, the Two-Stage RL is trained based on OptReward and OptGRPO. Through the data selection strategies, the second-stage RL improves sampling efficiency and the ability to solve complex problems. In contrast, many data-synthesis-based methods rely on high-quality datasets for SFT, which require larger data volumes and higher data quality while achieving limited performance.

We believe our updates have clarified the raised concerns. All changes have been highlighted in blue in the revised manuscript.

Due to the recent unforeseen incident, the reviewers are unable to provide further feedback or engage in additional discussion regarding my responses. We sincerely appreciate your effort in making the final decision.

Best regards,

The Authors

---

### Note · Authors · 2026-01-29

I have read and agree with the venue's withdrawal policy on behalf of myself and my co-authors.

---

### Meta-Review · Area_Chair_du9L · 2026-01-05

**Summary:**

Looking at the reviews and discussion, I'm recommending rejection. The paper presents a training pipeline for small optimization-solving LLMs, but the core issue is insufficient novelty relative to the established literature. Two reviewers scored it at 4 and one at 6, and while the authors provided helpful clarifications and additional experiments during rebuttal, these don't fully address the fundamental concerns about contribution.

The main problems are threefold. First, the motivation around "limited resources" is undermined by the use of 140K samples for SFT warm-up, which exceeds what related works like ORLM use. The authors argue this data is "low-cost" because it's public, but this doesn't change the fact that the resource requirements contradict the paper's premise. Second, the technical contributions feel incremental. The five-element modeling structure isn't novel, and OptGRPO's modifications to GRPO (removing KL divergence, adjusting clipping) are presented as design choices rather than principled algorithmic advances. The ablation studies show these help, but don't demonstrate why these specific modifications are the right ones beyond empirical tuning. Third, the framework's tight coupling to Pyomo limits its generalizability, and the lack of diagnostic metrics makes it hard to understand where the model actually fails.

The authors did add Step-OPT comparisons showing better parameter efficiency, which is valuable, but parameter reduction alone doesn't constitute a sufficient contribution when the methods achieving it aren't sufficiently novel. The reviewers consistently noted that while the empirical results are good, the paper reads more as solid engineering than a research contribution advancing our understanding of how to train LLMs for optimization. For a venue like ICLR, we need clearer technical or conceptual innovations beyond "we combined existing techniques and got a smaller model that works reasonably well."

**Reviewer Concerns:**

The rebuttal successfully addressed several concrete experimental gaps. The authors added Step-OPT comparisons across different model sizes and base models, demonstrating that MiniOpt maintains better performance when scaled down from 7B to 3B parameters compared to Step-OPT's degradation. They also included DAPO ablations showing a 2.37% performance drop when replacing OptGRPO, and cited ORLM results for comparison. The typo in the gradient clipping formula was corrected. These additions strengthen the empirical evaluation.

However, the core conceptual concerns remain outstanding. Reviewer Etef's fundamental concern about the "limited resources" claim was not resolved. The authors argue that 140K samples are "low-cost" because they're public and only trained for 1 epoch, but this sidesteps the issue that this still represents substantially more data than related work like ORLM. Saying the data acquisition cost is minimal doesn't change the data volume requirement. Similarly, while they showed SFT contributes less than RL to final performance, this doesn't explain why a method premised on resource constraints needs such extensive warm-up data in the first place.

The novelty concerns also persist. The authors clarified that their contribution is an efficient training pipeline rather than novel verification or algorithmic ideas, but this clarification actually reinforces rather than addresses the reviewers' concerns about limited technical innovation. Reviewer t9Ho noted that OptGRPO merely incorporates existing GRPO techniques, and while the authors explained their design choices (removing KL divergence, adjusting clipping bounds), they didn't provide principled justification for why these specific modifications are theoretically or empirically necessary beyond "they worked in our ablations."

Reviewer P5FB's concerns about Pyomo coupling and diagnostic metrics received explanations but no substantive changes. The authors described their solver selection strategy and suggested methods for error diagnosis, but acknowledged they don't actually provide fine-grained diagnostics in the paper. The framework remains tightly coupled to Pyomo, limiting generalizability claims.

**Reviewer Scores:**

Reviewer Etef (initial score: 4):

Likely would increase to 5, but still below acceptance. The Step-OPT comparisons at multiple scales and DAPO ablations directly addressed their experimental requests, which shows good faith engagement. However, their fundamental concerns about the "limited resources" contradiction and novelty of the problem setting were not resolved by the rebuttal. The authors' clarification that their contribution is the training pipeline itself actually confirms this is more engineering optimization than conceptual innovation. A reviewer this thorough and familiar with the literature would appreciate the added experiments but recognize the core issues remain.

Reviewer t9Ho (initial score: 4):

Could potentially increase to 5 or possibly 6. They received the Step-OPT and ORLM comparisons they requested, and the clarification about OptGRPO's role in the pipeline addresses some confusion. They initially gave positive remarks about the experimental design and practical metrics, so the additional experiments align with what they valued. However, their concerns about limited innovation in the verification scheme and GRPO modifications were acknowledged rather than refuted. Whether they'd reach 6 depends on how much they weight empirical contributions versus algorithmic novelty, but 5 seems more probable.

Reviewer P5FB (initial score: 6):

Would likely maintain 6, possibly dropping to 5. They were the most positive reviewer, emphasizing the strong empirical results and resource-conscious design. The rebuttal provided reasonable explanations for solver selection and error diagnosis, though without substantive methodological changes. The explanations about Pyomo coupling might actually highlight limitations they hadn't fully appreciated initially. Given their initial enthusiasm for the empirical work and comprehensive experiments, maintaining 6 is most likely, but the limitations discussion could prompt reconsideration to 5.

---

### Decision · Program_Chairs · 2026-01-26

Reject